# Social tolerance and role model diversity increase tool use learning opportunities across chimpanzee ontogeny
Oscar Nodé-Langlois[1,2,3] ✉, Eléonore Rolland[1,2,3,4], Cédric Girard-Buttoz [2,4,5], Liran Samuni[2,6],
Pier Francesco Ferrari[1], Roman M. Wittig[1,2,4] & Catherine Crockford[1,2,4] ✉

Social learning opportunities shape cognitive skills across species, especially in humans. Although the social environment impacts learning opportunities, the benefits of role model diversity and tolerance on task learning in tool-using species remain poorly understood. To explore these links, we study 2343 peering events (close-range observation of a conspecific) from 35 wild immature (<10 y) chimpanzees (*Pan troglodytes verus*). We find that chimpanzee peering functions to acquire information more than food, persists during development while peaking around weaning age, and increases with food processing complexity. Role models change throughout development, with increased peering at mothers during early stages and for more complex tasks. Finally, immatures observe many role models, favouring older and more tolerant individuals. We conclude that chimpanzees learn from multiple tolerant individuals, particularly when acquiring complex skills like tool use. Tolerant societies may be necessary for the acquisition and retention of the diverse tool kits rarely found in nature.

Social learning refers to learning that is influenced by observing or interacting with another individual or its products[1]. It is considered a key factor in the evolution and the ontogeny of technical skills, such as tool use[2]. In evolutionary and ontogenetic contexts, this learning strategy is particularly advantageous when the costs of individual learning are high because individual learning is dangerous or complex[3], particularly for tasks impossible to learn alone[4]. However, while social learning is widespread across taxa[5], tool use is only observed in a restricted group of species and is often stereotyped. The sparseness of this phenomenon across the animal kingdom suggests that a rare set of socio-ecological features has driven the evolution of versatile technical skill use: (1) the possibility to learn across a protracted developmental period[6], (2) access to tolerant caregivers[7], and (3) access to many role models[8]. Although these factors are likely essential, empirical evidence of how they influence social learning across contexts and developmental stages in social species remains scarce. Here, we present an empirical analysis that tests whether these three factors facilitate learning opportunities across ontogeny for a flexible tool user with protracted development, the western chimpanzee (*Pan troglodytes verus*).

Protracted development, in which reliance on parental care extends well beyond the weaning age, is considered an important factor for flexible

tool use and human evolution[6]. Various tool-using species, including capuchins, great apes, New Caledonian crows, elephants, and dolphins, display dependency on caregivers beyond weaning or fledging age[9–12]. Such protracted dependency likely aids in skill acquisition by extending social learning opportunities[13]. Supporting this hypothesis, learning complex technical skills often requires extended periods[14,15]. For instance, tool use proficiency in New Caledonian crows is reached up to 1 year after hatching[16] and in capuchins, after 4 years of age[17]. In orangutans and chimpanzees, some complex feeding skills continue to improve after the first decade of life[15,18]. We, therefore, expect that immatures of tool using species will continue to seek social learning opportunities until close to adulthood, with a preference for complex and unmastered tasks.

Social tolerance from role models is also considered a key factor for social learning, especially for complex skills[19]. In wild species, social tolerance influences learning opportunities. For instance, orangutans are more likely to observe spatially tolerant role models[20], while macaques are more likely to interact with grooming partners in tool use learning contexts[21]. High social tolerance facilitates close observation and may be necessary for certain forms of high-fidelity social learning, which can be particularly beneficial for transmitting complex skills in humans[22].

[1]Institut des Sciences Cognitives Marc Jeannerod, CNRS, Bron, France. [2]Taï Chimpanzee Project Centre Suisse de Recherches Scientifiques en Côte d'Ivoire, Abidjan, Côte d'Ivoire. [3]Université Claude Bernard Lyon 1, Villeurbanne, France. [4]Department of Human Behavior, Ecology and Culture, Max Planck Institute for Evolutionary Anthropology, Leipzig, Germany. [5]ENES Bioacoustics Research Laboratory Centre de Recherche en Neurosciences de Lyon, CNRS, Inserm, University of Saint-Etienne, Saint-Etienne, France. [6]Cooperative Evolution Lab, German Primate Center, Gottingen, Germany. ✉e-mail: oscar.node-langlois@isc.cnrs.fr; crockford@eva.mpg.de

Several studies suggest that social tolerance is beneficial for the acquisition of complex skills. Prosocial behaviours correlate with sequential problem-solving in children[23] and with the acquisition of composite tool use in captive chimpanzees[24]. In barbary macaques, adults are more likely to learn from huddling partners as task complexity increases[25]. Across species, caregivers demonstrate greater levels of social tolerance compared to other individuals and often serve as primary role models. In several species, including chimpanzees, caregivers' tolerance in tool use contexts provides opportunities for close observation[26], co-action[24], and individual practice through item transfers[10,27,28]. Evidence of caregiver influence on tool-use transmission[29,30], tool-use efficiency[28,31], and tool choice[16] have been reported in great apes, New Caledonian crows, and dolphins, suggesting that caregivers are influential role models for tasks involving tool use. However, the extent to which reliance on caregivers varies through ontogeny and with task complexity remains to be investigated. If caregiver tolerance facilitates learning, particularly for complex tasks, we expect immatures to seek learning opportunities from their mother early in development and for tasks that are difficult to master.

For learning technical skill diversity, access to not one but many tolerant role models is argued to be advantageous. In social species, social learning strategies may change through development. Infants and juveniles primarily learn from caregivers; however, as they mature, they also learn from other role models[32]. Access to alternative, tolerant, and experienced role models can significantly increase social learning opportunities. Several studies suggest larger group sizes enhance both the frequency and diversity of social learning opportunities[33], but see[34]. For instance, in Australian magpies, larger group sizes have been associated with increased emergence and spread of innovations[35]. In orangutans, the number of available role models and variation in sociality at the species level correlates with observational social learning opportunities[36]. Inter-species comparisons further suggest that social species tend to rely more on social learning when solving a multi-door task in jays[37], and when interacting with new food items in captive great-apes[38,39]. If role model diversity facilitates learning, we expect immatures of social species to learn not only from one but from many role models, with preferences for tolerant and more experienced individuals.

We tested these predictions in wild chimpanzees, one of the very few species known to master the use of large and diverse toolkits, exhibiting protracted development, high maternal dependency, and living in large social groups. Immature chimpanzees must acquire diverse and complex feeding strategies such as extractive foraging and tool use, some of which are acquired through social learning during development[40,41]. In the Taï National Park, chimpanzees feed on more than 500 different food items and regularly extract inaccessible nutritious foods, like nuts, insects, and honey, using diverse techniques[42]. These techniques can involve the use of tools such as stones, sticks, and leaves[43] used for a variety of actions[18]. While most feeding skills are acquired by weaning age[44], which typically occurs between 3 and 5 years[45], several skills involving tool use can take 10–15 years to master[18,46], and may involve sensitive learning periods[47,48]. Chimpanzees also have a protracted dependency and remain in association with their mothers for much of the first decade of life[49]. During this time, mothers may provide unique learning opportunities compared to other role models, particularly in complex tasks, such as nut-cracking, termite fishing, and ant dipping[46,50,51]. Chimpanzees live in large fission fusion societies, creating considerable variation in the availability of role models throughout each day and display a high level of social tolerance, which enables behaviours such as food transfers between unrelated individuals[52].

To evaluate the role of social learning during the acquisition of new behaviours, one approach is to use markers of social learning processes such as social visual attention[53]. Peering behaviour, defined as "directly looking at the action of another individual at a close enough range that enables the peering individual to observe the details of the action"[26], has been described in all great ape species[26,47,54–56] as well as in marmosets[57], white faced capuchins[9] and vervet-monkeys[58]. Peering behaviour requires sustained selective attention and likely provides opportunities for social learning. In wild orangutans, peering events are primarily produced during development and following dispersal[26,59], mostly in feeding and nest-building contexts—both behaviours thought to be socially influenced. Furthermore, peering in orangutans was followed by an increase in item manipulation and was more likely to occur when actions required for the processing of rare or difficult-to-process foods were performed, suggesting that peering facilitated learning. Several studies conducted on chimpanzees also suggest a link between peering behaviour and the acquisition of new behaviours[4,60–62]. The peering behaviour of immature wild chimpanzees has also been reported during the development of technical skills in the contexts of extractive foraging (*Saba florida* fruit)[63], nut cracking[47], termite-fishing[64], ant-dipping[51], and well digging[65]. While previous studies focused on peering in feeding contexts and for the acquisition of a limited number of skills, here we assessed peering across all observed contexts.

While a growing number of studies suggest a function of peering in social learning, evaluating its use across different contexts is necessary to disentangle learning from other proposed functions of peering. These include peering being a signal of dominance and an affiliative signal in captive bonobos[66]. Peering behaviour was highly correlated with relationship quality in captive bonobos but not in captive chimpanzees, suggesting that its function may differ between species[67]. In wild orangutans and chimpanzees, peering behaviour is often associated with explicit solicitations for food, suggesting that nutritional needs can drive this behaviour[26,63,68]. Explicit begging behaviours, such as extending a hand to another's food item or mouth or whimpering or screaming vocalisations, have been hypothesised to serve informational as well as nutritional needs[69–72]. However, they can also trigger beggar-directed aggression. In contrast, peering without explicit begging may allow individuals to closely observe complex tasks performed by those who might not tolerate explicit solicitations. Therefore, we expect nutritional and informational benefits to differ between peering and explicit begging, with peering providing a greater opportunity to observe processing steps, while explicit begging may better facilitate food acquisition.

To understand the use of peering and identify from whom chimpanzees learn, we developed four sets of models to test the following four hypotheses:

1. *Peering functions to acquire information rather than food.* We predicted that: (1a) Peering alone will elicit less food transfers than explicit begging. (1b) As social learning usually involves an alternation of observation and individual practice, peering in tool use contexts will induce a higher probability of manipulating tools than explicit begging. (1c) Processing complexity will increase the relative use of peering compared to explicit begging.

2. *Peering continues across protracted development, more for complex tasks.* Here, we predicted that peering will be prevalent when individuals can socially learn food processing skills, specifically: (2a) The frequency of peering events in foraging contexts will peak around weaning age (between 3 and 5 years), when most of the feeding skills are being acquired[44,45,73], but peering behaviour will persist later in development. (2b) Food items that are difficult to process or access will elicit a higher frequency of peering events. (2c) Failures in one's extractive feeding attempts will subsequently increase the probability of peering.

3. *Early learning and learning of complex tasks rely on mothers.* As mothers were more spatially tolerant than other individuals and might provide favourable learning opportunities, we expect offspring to peer preferentially at their mother early in development and for complex tasks. We therefore predicted that: 3a) The proportion of peering directed at the mother will decrease with age as immatures acquire their own skills and other models become preferred. 3b) Immatures will preferentially look at their mothers over alternative role models for the acquisition of complex feeding skills.

4. *Learning is facilitated by many tolerant group members.* To increase social learning opportunities, we expect immatures to target a high diversity of role models, especially around weaning age. If peering at non-mother role models requires social tolerance and is used to gain

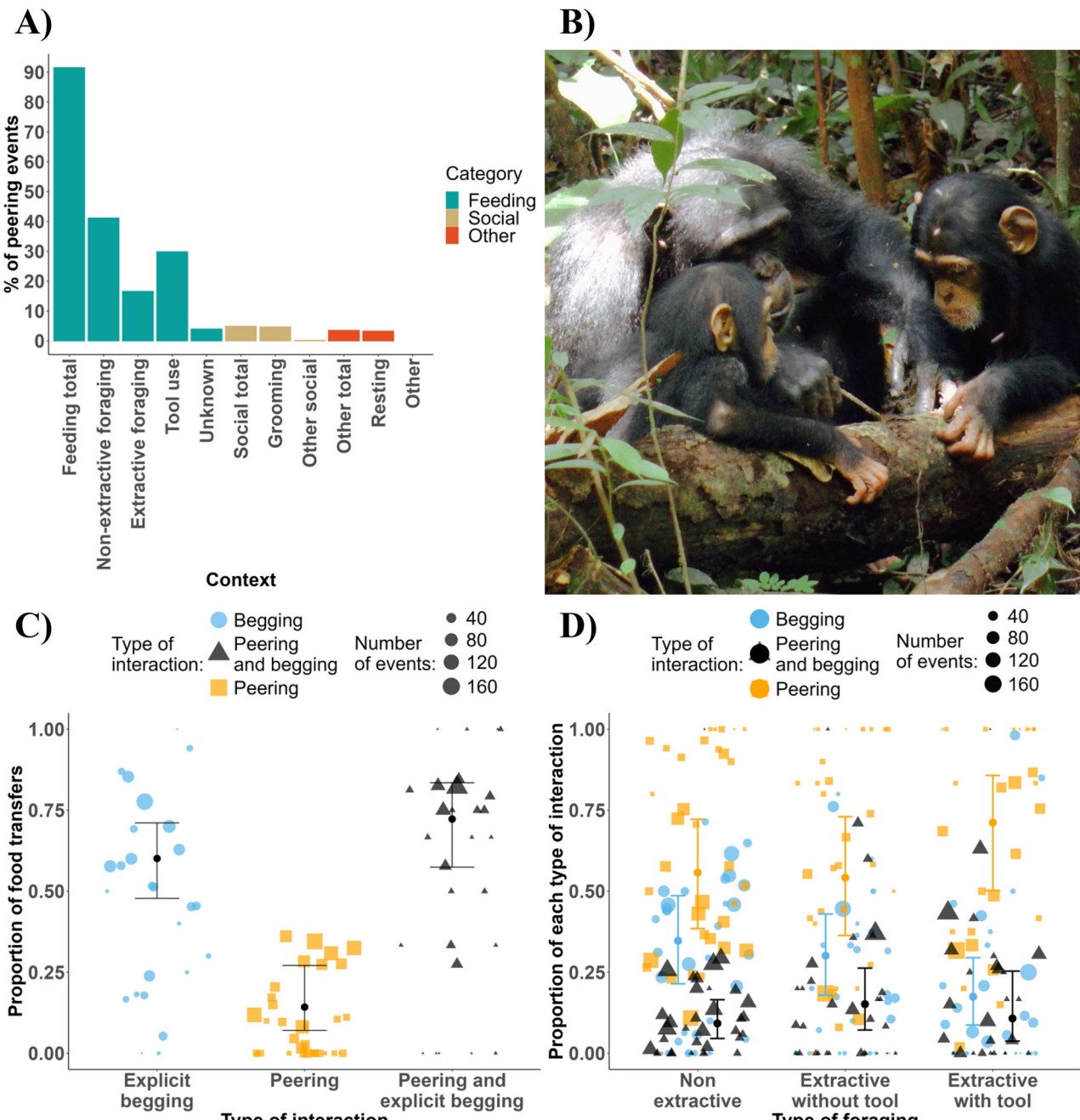

**Fig. 1 | Results indicating the function of peering compared to explicit begging.**
**A** Proportion of peering events produced by immature chimpanzees per context.
**B** Adult female extracting larvae with a stick while her offspring and an unrelated immature are peering at her actions. Credit: Liran Samuni/Taï Chimpanzee Project.
**C** Effect of social interaction type on the likelihood of obtaining food (model 1a).

**D** Effect of food processing type on the proportion of social interactions that are, or include, peering events (model 1c). The dots with the vertical line indicate the estimate and the 95% credible interval from the Bayesian Regression model. Each dot represents the proportion of food transfers per subject (**C**) and the proportion of each type of social interaction per subject (**D**).

valuable information, we expect selectivity for knowledgeable (i.e., older than themselves or adults) and tolerant role models. We, therefore, predicted that: (4a) The number of observed role models will peak around weaning age. (4b) Peerers will preferentially seek role models with whom they can comfortably maintain close proximity, and who are adult or older than themselves.

## Results
### Peering behaviour in immature chimpanzees
We recorded a total of 2343 peering events of chimpanzees younger than 10 years. Overall, 92% of these events occurred in foraging contexts (Fig. 1A; for

instance, see Fig. 1B) followed by social contexts (5%). Most of the peering events observed in social contexts occurred during grooming ($n = 112$, 4.8%).

**Model 1a: Peering behaviour will elicit less food transfers than explicit begging behaviour.** Supporting our prediction, we found that peering events alone were consistently less likely to be followed by food transfers (estimate: −1.67, 95% CI: [−2.48; −0.85], see Fig. 1C, Table 1) compared to explicit begging. However, peering coupled with explicit solicitations was associated with a higher probability of food transfers than explicit begging alone (estimate: 0.86, 95% CI:

**Article**

## Table 1 | Models testing the function of peering

| Prediction | Response | Term | Estimate | SE | 95% CI | 95% CI | 89% CI | 89% CI |
|---|---|---|---|---|---|---|---|---|
| Model 1a) Peering will elicit less food transfers than explicit begging behaviour | Food transfers following peering, begging or both in foraging context (yes or no) | Intercept | −1.49 | 0.66 | −2.79 | −0.20 | −2.52 | −0.43 |
| | | Age | 0.33 | 0.24 | −0.16 | 0.79 | −0.07 | 0.70 |
| | | **Age$^2$** | **−0.73** | **0.20** | **−1.15** | **−0.38** | **−1.06** | **−0.45** |
| | | Sex (Male) | 0.29 | 0.39 | −0.48 | 1.07 | −0.34 | 0.92 |
| | | Maternal rank | −0.28 | 0.23 | −0.72 | 0.17 | −0.63 | 0.09 |
| | | Group North[a] | 0.28 | 0.56 | −0.82 | 1.35 | −0.62 | 1.18 |
| | | Group South[a] | 0.11 | 0.43 | −0.72 | 0.95 | −0.56 | 0.80 |
| | | **Receiver (Mother)[d]** | **2.33** | **0.40** | **1.56** | **3.14** | **1.70** | **2.97** |
| | | **Receiver (Other)[d]** | **−0.94** | **0.45** | **−1.82** | **−0.06** | **−1.65** | **−0.22** |
| | | Non extractive[b] | −0.21 | 0.31 | −0.8 | 0.40 | −0.69 | 0.28 |
| | | Extractive with tool[b] | 0.14 | 0.35 | −0.54 | 0.82 | −0.41 | 0.68 |
| | | **Peering[c]** | **−1.67** | **0.41** | **−2.48** | **−0.85** | **−2.31** | **−1.01** |
| | | **Peering + Explicit begging[c]** | **0.86** | **0.39** | **0.10** | **1.67** | **0.25** | **1.50** |
| | | Monopolizability (High) | −0.23 | 0.37 | −0.95 | 0.51 | −0.80 | 0.35 |
| | | *Non extractive[c]:Peering[c]* | *−0.59* | *0.37* | *−1.30* | *0.15* | *−1.17* | *0.00* |
| | | **Extractive with tool[b]:Peering[c]** | **0.74** | **0.38** | **0.00** | **1.48** | **0.14** | **1.34** |
| | | Non extractive[b]:Peering + Explicit begging[c] | −0.47 | 0.35 | −1.16 | 0.21 | −1.03 | 0.09 |
| | | Extractive with tool[b]:Peering + Explicit begging[c] | 0.01 | 0.39 | −0.77 | 0.79 | −0.62 | 0.64 |
| Model 1b) Peering in tool use contexts will induce a higher probability of manipulating tools than explicit begging behaviour | Latency before the manipulation of a tool following peering or explicit begging in tool use context | Intercept | −1.76 | 0.68 | −3.10 | −0.40 | −2.82 | −0.67 |
| | | Age | −0.25 | 0.21 | −0.66 | 0.15 | −0.59 | 0.07 |
| | | Receiver (Mother) | 0.42 | 0.35 | −0.25 | 1.14 | −0.12 | 0.98 |
| | | Sex (Male) | 0.17 | 0.52 | −0.88 | 1.19 | −0.66 | 1.00 |
| | | Party size | 0.18 | 0.20 | −0.19 | 0.59 | −0.12 | 0.50 |
| | | Group North[a] | 0.66 | 0.73 | −0.80 | 2.04 | −0.50 | 1.80 |
| | | Group South[a] | −0.08 | 0.58 | −1.23 | 1.05 | −1.01 | 0.84 |
| | | Peering + Explicit begging | −0.14 | 0.37 | −0.87 | 0.60 | −0.73 | 0.45 |
| | | Peering[c] | −0.12 | 0.37 | −0.84 | 0.65 | −0.71 | 0.48 |
| | | Food transfer (Yes) | −0.15 | 0.24 | −0.64 | 0.34 | −0.54 | 0.24 |
| Model 1c) Food processing complexity will increase the relative use of peering compared to explicit begging behaviour | Type of social interaction (Peering) with explicit begging as reference level | Intercept | 1.41 | 0.72 | 0.00 | 2.87 | 0.28 | 2.59 |
| | | Age | 0.39 | 0.32 | −0.29 | 0.99 | −0.13 | 0.88 |
| | | *Age$^2$* | *0.35* | *0.22* | *−0.08* | *0.80* | *0.01* | *0.70* |
| | | Sex (Male) | −0.64 | 0.48 | −1.56 | 0.31 | −1.40 | 0.14 |
| | | **Receiver (Mother)[d]** | **−2.27** | **0.31** | **−2.90** | **−1.67** | **−2.77** | **−1.79** |
| | | *Receiver (Other)[d]* | *0.63* | *0.33* | *−0.03* | *1.26* | *0.11* | *1.15* |
| | | *Maternal rank* | *0.44* | *0.27* | *−0.08* | *0.96* | *0.01* | *0.86* |
| | | Non extractive[b] | −0.16 | 0.29 | −0.73 | 0.40 | −0.62 | 0.30 |
| | | **Extractive with tools[b]** | **0.91** | **0.33** | **0.24** | **1.56** | **0.37** | **1.44** |
| | | Group North[a] | 0.18 | 0.67 | −1.18 | 1.49 | −0.90 | 1.24 |
| | | Group South[a] | −0.07 | 0.51 | −1.05 | 0.93 | −0.87 | 0.74 |
| | Type of social interaction (Peering) with explicit begging + Peering as reference level | Intercept | −0.76 | 0.71 | −2.17 | 0.59 | −1.90 | 0.35 |
| | | Age | −0.03 | 0.32 | −0.70 | 0.61 | −0.55 | 0.48 |
| | | *Age$^2$* | −0.03 | 0.20 | −0.46 | 0.33 | −0.36 | 0.26 |
| | | Sex (Male) | 0.33 | 0.44 | −0.57 | 1.20 | −0.37 | 1.02 |
| | | Receiver (Mother)[d] | −0.03 | 0.40 | −0.80 | 0.77 | −0.66 | 0.64 |
| | | Receiver (Other)[d] | −0.36 | 0.45 | −1.21 | 0.52 | −1.06 | 0.37 |
| | | Maternal rank | 0.21 | 0.26 | −0.26 | 0.72 | −0.18 | 0.63 |
| | | **Non extractive[b]** | **−0.63** | **0.26** | **−1.13** | **−0.11** | **−1.05** | **−0.21** |
| | | Extractive with tools[b] | 0.21 | 0.41 | −0.61 | 1.01 | −0.46 | 0.85 |
| | | Group North[a] | −0.10 | 0.65 | −1.30 | 1.20 | −1.12 | 0.94 |
| | | Group South[a] | −0.07 | 0.51 | −1.05 | 0.93 | −0.87 | 0.74 |

Results of our statistical models 1a, 1b, and 1c comparing the outcomes and contexts of peering and explicit begging, including the dependent variables, the effects with associated estimated error and credible interval at 95 and 89% (respectively in bold or italic if they did not cross 0).
[a]Group East.
[b]Extractive foraging without tools.
[c]Explicit begging.
[d]Maternal kin as reference categories, other contrasts are shown on (Table S3).

[0.10; 1.67], see Fig. 1C, Table 1). The two-way interaction between the modality of social interaction (peering, explicit begging or both) and the type of food processing, showed that the probability of food transfers increased with task complexity following peering events but not following explicit begging alone (estimate: 0.74, 95% CI: [0.00; 1.48], see Fig. S2B, Table 1). We also found an effect of kinship, with mothers and siblings being more likely to share food. Mothers were more likely than siblings to share food (estimate: 2.33, 95% CI: [1.56; 3.14], see Fig. S2C, Table 1), who were more likely to share food than other individuals (estimate: 0.94, 95% CI: [0.06; 1.82], see Fig. S2C, Table 1).

### Model 1b: Peering behaviour in tool use contexts will reduce the latency before manipulating tools more than explicit begging behaviour.
Contrary to our prediction, we found no evidence that peering compared to explicit begging in tool use contexts reduces the latency to manipulate tools, with 89% CI overlapping zero.

### Model 1c: Foraging complexity will increase the relative use of peering behaviour compared to explicit begging behaviour.
Supporting our prediction, we found that extractive foraging with tools, compared to extractive foraging without tools (estimate: 0.91, 95% CI: [0.24; 1.56], see Fig. 1D, Table 1) and non-extractive foraging contexts (estimate: 1.07, 95% CI: [0.35; 1.79], see Fig. 1D, Table S3) was associated with an increased proportion of peering compared to explicit begging. The proportion of social interactions that included peering increased from 80% in non-extractive contexts to 90% in tool-use contexts. Additionally, we found that extractive foraging contexts without tools (estimate: 0.63, 95% CI: [0.11; 1.13], see Fig. 1D, Table 1) and with tools (estimate: 0.85, 85% CI: [0.15; 1.53], see Fig. 1D, Table S3), compared with non-extractive foraging contexts, were associated with an increased proportion of social interactions including both peering and explicit begging compared to explicit begging alone.

Model 1c also indicated an effect of kinship, with mothers and siblings being proportionally more likely than other individuals to receive explicit begging than peering alone, see Fig. S3A. Mothers received a higher proportion of begging than other maternal kin (estimate: 2.27, 95% CI: [1.67; 2.90], see Fig. S3A, Table 1), who received a higher proportion of begging than other individuals (estimate: 0.63, 89% CI: [0.11; 1.15], Table1). Model 1c also indicated a moderated positive effect of maternal rank on the proportion of peering (estimate: 0.44, 89% CI: [0.01; 0.86], Table 1).

### Peering continues across protracted development, more for complex tasks
Here, we tested the prediction that peering behaviour will peak around weaning age but will persist later in development and increase for complex and unmastered tasks. Our model results supported these predictions:

**2a)** The frequency of peering events peaked during development between 4 and 5 years old (Fig. 2A) (estimate: −0.41, 95% CI: [−0.87; −0.03], Table 2), when most feeding skills are learned[44].

**2b)** Food processing complexity and resource monopolizability increased the use of peering behaviour (model 2b, resource monopolizability—estimate: 0.85, 95% CI: [0.37; 1.30], see Fig. 2B, Table 2, extractive foraging, both with and without tools compared to non-extractive (estimate: 1.15, 95% CI: [0.16; 2.08], estimate: 0.56, 95% CI: [0.12; 0.95], see Fig. 2C, Table 2). The peering rate was three times higher in extractive foraging with tools compared to non-extractive foraging.

**2c)** Food extraction failure led to a shorter subsequent latency to peer at role models than success (model 2c, estimate: 1.17, 95% CI: [0.64; 1.73], see Fig. 2D, Table 2), with the effect diminishing over time. After 10 min, the predicted probability of having peered was 50% for failures versus 20% for successes, increasing to 60% versus 30% at 20 min and 75% versus 45% at 40 min.

### Peering during early development and at complex tasks is directed at mothers
Here, we tested the prediction that offspring are more likely to peer at their mother early in development and for more complex tasks. We found support for our prediction:

**3a)** Immatures exhibited a shift from observing their mother to observing other individuals as their age increased (estimate: -0.73, 95% CI: [-1.19; -0.24], see Fig. 3A, Table 3). Mothers were observed in more than 80% of the cases during the 1st year and this decreased to less than 20% after 9 years.

**3b)** Task complexity influenced role model choices such that in larger parties, immatures were more likely to peer at other models than the mother during non-extractive foraging, but not during extractive foraging without tools (estimate: −0.81, 95% CI: [−1.44; −0.21], see Fig. 3B, Table 3) and extractive foraging with tools (estimate: −0.76, 95% CI: [−1.40; −0.13], see Fig. 3B, Table S3). Model 3b also weakly supported an overall positive effect of tool use on the probability of peering at the mother compared to non-extractive foraging (estimate: 1.15, 89% CI: [0.16; 2.08], see Fig. 3B, Table S3).

### Peering is directed at many tolerant role models
Here, we tested the prediction that immatures will rely on varied non-mother role models during development, especially when learning is most needed, with a bias toward older and spatially tolerant role models. We found support for our prediction:

**4a)** The number of observed role models peaked close to weaning age at 12 individuals observed per 60 h (estimate: -0.16, 95% CI: [-0.27; -0.00], see Fig. 4A, Table 4), between 5 and 6 years and decreased after. Note, even 9-year-olds continue to peer at a mean of 7 role models per 60 h.

**4b)** Examining how social tolerance and partner identity impacts peering probability (model 4b), we found that immatures were more likely to peer at tolerant individuals: those with whom they spent more time in less than 1 m proximity (estimate: 0.35, 95% CI: [0.21; 0.65], see Fig. 4C, Table 4). Individuals spending close to 0% of their association time in close proximity were three times less likely to be peered at than individuals spending 10% of their association in proximity to the peerer.

We also found that older role models were more likely to be peered at (estimate: 0.68, 95% CI: [0.10; 0.76]) and that this effect increased with the age of the focal (estimate: 0.27, 95% CI: [0.07; 0.47], see Fig. 4B, Table 4).

### Discussion
Our findings strongly support the idea that chimpanzees seek social learning opportunities at least for the first 10 years of their development, mostly around weaning age and for complex and unmastered skills. Immatures mainly relied on their mother, especially for skills difficult to learn, such as those involving tool use. Nonetheless, individuals other than mothers or maternal kin also served as role models, particularly for older immatures and in relationships characterized by increased tolerance toward the immature. In species with advanced technical skills, the existence of large, tolerant societies may be essential for retaining an extensive and varied tool kit, which is common in some chimpanzee populations but is extremely rare in the animal kingdom outside of humans.

We observed that 92% of peering events occurred during feeding, compared with only 5% during social interactions outside of feeding contexts. Many of these peering events (N > 846) were directed at specific feeding behaviours for which social learning has previously been implicated (studies demonstrating community or population differences indicative of cultural variation), such as tool use[47,74,75], prey preferences[76], a nd some types of fruit processing[77]. Learning detailed food processing procedures may require the close examination offered by peering. In contrast, the social learning of dynamic social behaviours such as play or agonism does not lend itself to close, passive observation and hence may require learning through direct social engagement. An exception may be more static grooming interactions, which accounted for most peering events in social contexts. Our results mirror observations from other species, notably orangutans[26]

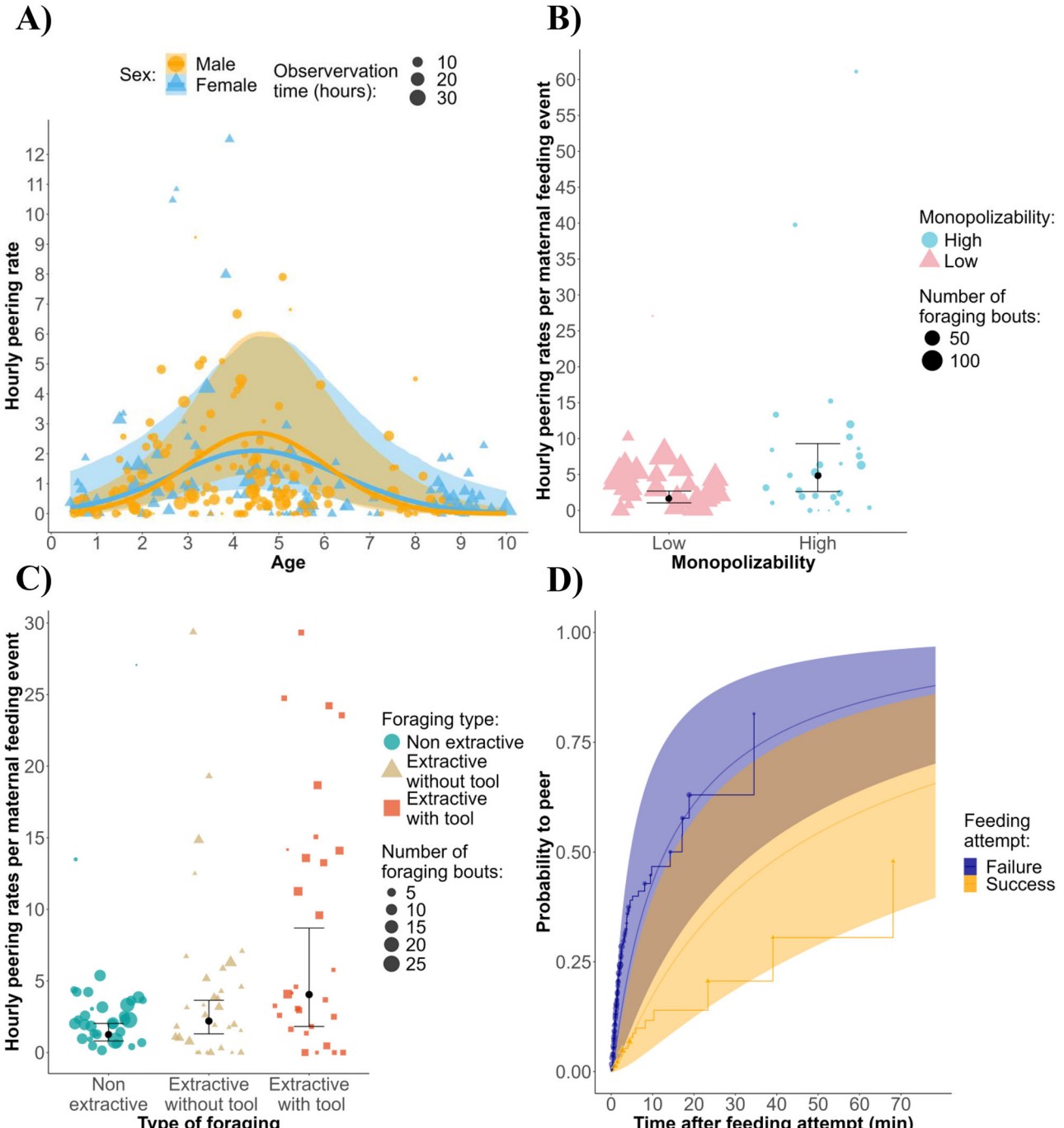

**Fig. 2 | Peering behaviour during the acquisition of feeding skills. A** Predicted effect of age and sex on the total rate of peering events (model 2a). **B** Effect of food monopolizability on the rate of peering events while the mother is feeding (model 2b). **C** Effect of foraging type on the rate of peering events while the mother is feeding. **D** Effect of immature feeding success on the probability to peer over time

(model 2c). The regression line from the Bayesian Regression model shows a 95% credible interval. The dots with the vertical line indicate the estimate and the 95% credible interval from the Bayesian Regression model. Each dot represents the observation time per subject and month (**A**), the number of foraging bouts for the shown type of processing per subject (**B, C**), and the number of peering events (**D**).

and humans[78], in which observational learning is mostly used for technical skills.

Additionally, peering events were three times less likely to result in food transfers than explicit begging, supporting the hypothesis that peering is less driven by nutritional needs (model 1a) and also occurred in contexts during which food was easily accessible (personal observation), suggesting additional motivation other than nutritional needs. Compared to explicit begging, peering was more likely to occur in tool use contexts, supporting the hypothesis that peering is used to gain information. Nonetheless, peering might still facilitate food acquisition

given that peering coupled with explicit begging was associated with a higher probability of food transfers (see Fig. 1C) and access to occasional food remnants around tool use sites during complex feeding tasks could reduce the need for explicit begging to obtain food (see Fig. S2B). Additionally, peering did not decrease the latency until the next tool manipulation attempt (model 1b) compared to explicit begging, possibly because motivation to feed also promotes tool manipulation[79].

Together, these findings contribute to the growing body of evidence that peering provides social learning opportunities and might be used to estimate the prevalence of social learning for technical skill acquisition in

**Table 2 | Models testing the use of peering with respect to peerer's age and success, and the food processing context**

| Prediction | Response | Term | Estimate | SE | 95% CI | 95% CI | 89% CI | 89% CI |
|---|---|---|---|---|---|---|---|---|
| Model 2a) The frequency of peering events in foraging contexts will peak during development and persist after weaning | Number of peering events | Intercept | −0.05 | 0.44 | −0.92 | 0.83 | −0.75 | 0.66 |
| | | Sex (Male) | 0.33 | 0.40 | −0.44 | 1.12 | −0.30 | 0.97 |
| | | Age | −0.09 | 0.27 | −0.66 | 0.42 | −0.53 | 0.31 |
| | | **Age$^2$** | **−0.41** | **0.21** | **−0.87** | **−0.03** | **−0.75** | **−0.09** |
| | | **Party size** | **0.19** | **0.09** | **0.01** | **0.37** | **0.05** | **0.34** |
| | | Group North[a] | −0.11 | 0.49 | −1.07 | 0.84 | −0.89 | 0.66 |
| | | *Group South[a]* | *0.69* | *0.37* | *−0.06* | *1.43* | *0.07* | *1.28* |
| | | Maternal rank | 0.13 | 0.16 | −0.17 | 0.46 | −0.12 | 0.39 |
| | | Sex (Male): Age | 0.12 | 0.35 | −0.57 | 0.83 | −0.43 | 0.68 |
| | | *Sex (Male): Age$^2$* | *−0.48* | *0.29* | *−1.06* | *0.11* | *−0.95* | *−0.02* |
| Model 2b) Increasing food processing complexity will elicit a higher frequency of peering events | Number of peering events when at least one adult (the mother) is foraging | Intercept | 0.39 | 0.45 | −0.50 | 1.29 | −0.33 | 1.11 |
| | | *Age* | *−0.37* | *0.24* | *−0.88* | *0.08* | *−0.77* | *0.00* |
| | | **Non extractive[b]** | **−0.56** | **0.21** | **−0.95** | **−0.12** | **−0.88** | **−0.20** |
| | | *Extractive with tools[b]* | *0.61* | *0.37* | *−0.13* | *1.30* | *0.02* | *1.18* |
| | | Sex (Male) | 0.12 | 0.33 | −0.53 | 0.78 | −0.40 | 0.64 |
| | | *Age$^2$* | *−0.35* | *0.21* | *−0.81* | *0.02* | *−0.70* | *−0.04* |
| | | **Monopolizability (High)** | **0.85** | **0.23** | **0.37** | **1.30** | **0.46** | **1.21** |
| | | Group North[a] | −0.31 | 0.52 | −1.34 | 0.74 | −1.12 | 0.52 |
| | | Group South[a] | 0.48 | 0.38 | −0.26 | 1.26 | −0.12 | 1.09 |
| | | Maternal rank | −0.18 | 0.18 | −0.54 | 0.20 | −0.47 | 0.12 |
| | | Party size | 0.10 | 0.10 | −0.10 | 0.30 | −0.06 | 0.26 |
| | | Non extractive[b]:Age | 0.20 | 0.17 | −0.12 | 0.54 | −0.06 | 0.47 |
| | | Extractive with tool[b]:Age | −0.12 | 0.31 | −0.71 | 0.52 | −0.59 | 0.39 |
| | | Non extractive[b]: Age$^2$ | 0.08 | 0.14 | −0.19 | 0.34 | −0.14 | 0.30 |
| | | Extractive with tool[b]: Age$^2$ | −0.03 | 0.22 | −0.47 | 0.39 | −0.39 | 0.31 |
| Model 2c) Failure in feeding attempt will subsequently increase the probability of peering | Latency before a peering event following a feeding attempt | Intercept | −1.76 | 0.75 | −3.22 | −0.25 | −2.95 | −0.54 |
| | | **Success (Yes)** | **1.17** | **0.27** | **0.64** | **1.73** | **0.74** | **1.61** |
| | | Food type (Pod opening) | 0.42 | 0.28 | −0.14 | 0.99 | −0.04 | 0.89 |
| | | *Age* | *0.51* | *0.33* | *−0.11* | *1.18* | *0.00* | *1.05* |
| | | Sex (Male) | −0.11 | 0.57 | −1.22 | 1.04 | −0.99 | 0.80 |
| | | Age$^2$ | 0.31 | 0.25 | −0.14 | 0.84 | −0.07 | 0.74 |
| | | *Party size* | *0.28* | *0.18* | *−0.06* | *0.65* | *0.01* | *0.57* |
| | | Group North[a] | 0.29 | 0.79 | −1.29 | 1.85 | −0.95 | 1.59 |
| | | Group South[a] | 0.14 | 0.61 | −1.09 | 1.32 | −0.86 | 1.09 |
| | | Maternal rank | −0.09 | 0.36 | −0.82 | 0.63 | −0.67 | 0.48 |

Results of our statistical models 1a, 1b, and 1c, testing for the effect of contexts on peering, including the dependent variables, the effects with associated estimated error and credible interval at 95 and 89% (respectively in bold or italic if they did not cross 0).
[a]Group East.
[b]Extractive foraging without tools as reference categories, other contrasts are shown on (Table S3).

some wild species[53,80], although further research is needed to directly demonstrate the effects of peering behaviour on skill acquisition.

Chimpanzee peering peaked between 4 and 5 years old, corresponding to weaning age—an intensive period of feeding skills acquisition[44], and potentially a sensitive period to acquire nut cracking[47,48]. Although less frequent, peering behaviour did not disappear after weaning but persisted until at least 10 years old (the oldest individual in our sample), after less complex, but before more complex extractive, foraging skills have been honed. Skills required for successful nut-cracking and stick tool use for hidden foods, for example, continue to improve beyond the first decade of life[18,46]. Immature chimpanzees did not peer with similar rates at all feeding behaviours. Rather, peering was more often directed at individuals performing more complex tasks requiring extractive foraging. Peering behaviour was not only modulated by task complexity but also by the technical skills of immatures. Unsuccessful extractive foraging attempts doubled the probability of immatures to peer within 15 min after failure, similar to reports from captive chimpanzees[62]. Food item monopolizability also increased the likelihood of peering, consistent with studies suggesting that situations constraining learning opportunities, such as feeding competition[81] or resource scarcity[26] increase reliance on social learning. These findings support the idea that peering behaviour is employed in more challenging situations, for example, when individual learning is difficult or restricted. Peering

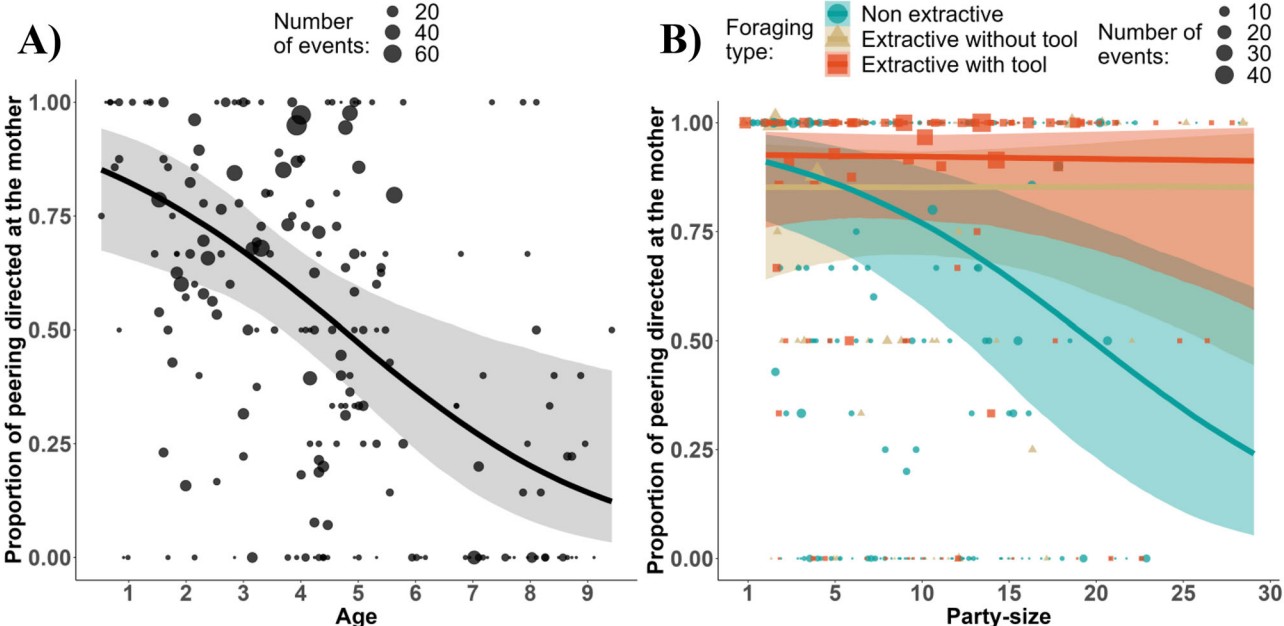

**Fig. 3 | Role model choices as developmental stage, food processing complexity, and party size change. A** Effect of age on the probability of peering at other individuals than the mother (model 3a). **B** Interaction between party size and food processing complexity on the probability of peering at other individuals while at least

one individual (the mother) is foraging (model 3b). The regression lines from the Bayesian Regression model show a 95% credible interval. Each dot represents the number of peering events per subject and age in a month: (**A**), per individual, type of foraging, and party size (**B**).

events in our data occurred from young ages (<2 year), before immatures reach the required physical and cognitive skills to solve tasks like tool use[82], suggesting that despite likely playing an important role in social learning, peering during development might not always lead to an immediate improvement in a specific skill. In children, for instance, the benefits of tool use demonstrations are constrained by the child's age and the number of observations[83]. Together, our results support the idea that protracted dependency provides greater opportunities for social learning, particularly for complex skills such as tool use.

Mothers are likely the best facilitators of their offspring's learning, allowing close observations and transfers of tools and food items[27,30,31,46,51,84]. Indeed, our results support the idea that mothers are more tolerant than other community members as immatures spent more time in close proximity to their mothers than to other individuals throughout their first decade of life (Fig. S4). Peering and explicit begging directed at the mother, rather than others, were also followed by a higher probability of food transfers (Fig. S2C). However, as offspring aged, we observed a gradual shift from peering primarily at the mother to peering more at other role models. This transition is evident in various mammal species[26,58], and notably, humans where time spent in association with the mother[85] and trust in her expertise decrease during development[86,87]. Nonetheless, throughout development, chimpanzee mothers were more likely to be used as role models, especially for feeding skills involving tool use, than other individuals. Party size had a smaller effect on the probability of peering at other individuals in extractive foraging compared to non-extractive foraging contexts. This suggests that task complexity biases offspring attention toward the mother and emphasises the strong influence of chimpanzee mothers on their offspring's tool-use behaviours despite the availability of other role models.

This continued importance of mothers, particularly for complex task learning years after weaning, may partly explain why immatures remain closely associated with their mothers at least through the first decade of life[49]. It remains to be tested whether the need to learn tool use techniques for extracting high-nutrient foods, such as nuts and insects, which can take

more than a decade to master[18,46], contributed to the evolution of hominid prolonged juvenile dependency. It is likewise unknown whether relying on tool-use learning from their mothers contributes to the stress levels, reduced physical growth, and loss of reproductive success that chimpanzees experience when orphaned before adulthood - possibly before they have acquired full tool kit skills[45,88–90]. Additionally, such patterns of social learning might be essential for the emergence of teaching, as associated costs are more likely to be compensated if they can increase the tutors' inclusive fitness and are restricted to the transmission of complex skills[7]. In chimpanzees and in orangutans (*Pongo abelii*), studies suggest that mothers facilitate access to food and tools depending on task complexity[50,69]. However, outside of humans, explicit teaching-like interactions are rare, though they have been reported in chimpanzee mother-offspring dyads during nut-cracking[46].

In addition to their mothers, immatures peered at many other group members, particularly around weaning age—12 different individuals per 60 h. Other individuals were feeding 27% of the time that the mother was not feeding, which might substantially increase learning opportunities. Our results also suggest that party size increases the frequency of peering (see model 2a Table 2). This implies that immature chimpanzees can potentially learn from a large part of their community and supports that group size, associated with conditions enabling access to role models such as social tolerance, might be key factors for the maintenance and the diversification of cultural dynamics[33].

Given that peering requires close proximity to role models, social tolerance is a prerequisite for peering, especially toward adults who might represent a potential threat. Our results support this idea as mature individuals who spent more time in 1 m proximity to the focal were also the most likely to be peered at. While a preference for mature individuals, presumably those who possess better skills[3], is also observed in capuchins and macaques[9,21], this contrasts with western gorillas, in which adults are less likely to be peered at than in chimpanzees, potentially because of a lower prosociality[54]. The preference for adult role models increased during development, after most of the basic skills had been acquired. These results are consistent with a chimpanzee nut cracking field experiment[47] and suggest

**Table 3 | Models testing the effect of developmental stage and food processing complexity on role model choices**

| Model | Response | Term | Estimate | SE | 95% CI | 95% CI | 89% CI | 89% CI |
|---|---|---|---|---|---|---|---|---|
| Model 3a) The proportion of peering directed at the mother will decrease with age | Identity of the receiver (mother or other individuals) | Intercept | 0.25 | 0.52 | −0.79 | 1.26 | −0.59 | 1.07 |
| | | **Age** | **−0.73** | **0.24** | **−1.19** | **−0.24** | **−1.10** | **−0.35** |
| | | Party size | −0.12 | 0.12 | −0.37 | 0.12 | −0.33 | 0.07 |
| | | Sex (Male) | −0.02 | 0.39 | −0.81 | 0.73 | −0.66 | 0.61 |
| | | Group North[a] | −0.17 | 0.57 | −1.29 | 0.98 | −1.08 | 0.74 |
| | | Group South[a] | 0.08 | 0.45 | −0.80 | 0.96 | −0.63 | 0.81 |
| | | Presence of maternal kin (Yes) | 0.05 | 0.32 | −0.58 | 0.70 | −0.47 | 0.56 |
| | | Maternal rank | 0.34 | 0.22 | −0.75 | 0.12 | −0.68 | 0.03 |
| | | Age:Party size | 0.10 | 0.11 | −0.31 | 0.11 | −0.27 | 0.07 |
| Model 3b) Immatures will preferentially look at their mothers over alternative role models for the acquisition of complex feeding skills | Identity of the receiver (mother or other individuals) when at least one adult (the mother) is feeding | Intercept | 1.92 | 0.86 | 0.24 | 3.61 | 0.53 | 3.27 |
| | | Age | −0.41 | 0.42 | −1.22 | 0.43 | −1.07 | 0.27 |
| | | Non extractive[b] | −0.44 | 0.44 | −1.30 | 0.47 | −1.14 | 0.25 |
| | | Extractive with tools[b] | 0.71 | 0.53 | −0.37 | 1.73 | −0.15 | 1.55 |
| | | Party size | 0.01 | 0.29 | −0.56 | 0.59 | −0.46 | 0.47 |
| | | Sex (Male) | 0.50 | 0.61 | −0.70 | 1.69 | −0.48 | 1.47 |
| | | Monopolizability (High) | 0.10 | 0.60 | −1.09 | 1.31 | −0.84 | 1.07 |
| | | Group North[a] | −0.36 | 0.80 | −1.89 | 1.21 | −1.61 | 0.92 |
| | | Group South[a] | −0.21 | 0.67 | −1.48 | 1.10 | −1.27 | 0.87 |
| | | Maternal rank | 0.04 | 0.37 | −0.65 | 0.81 | −0.52 | 0.65 |
| | | Presence of maternal kin (Yes) | −0.67 | 0.53 | −1.74 | 0.38 | −1.52 | 0.18 |
| | | Non extractive[b]:Age | −0.11 | 0.34 | −0.79 | 0.54 | −0.67 | 0.43 |
| | | Extractive with tool[b]:Age | −0.37 | 0.46 | −1.29 | 0.50 | −1.10 | 0.35 |
| | | **Non extractive[b]:Party size** | **−0.81** | **0.32** | **−1.44** | **−0.21** | **−1.32** | **−0.31** |
| | | Extractive with tool[b]: Party size | −0.05 | 0.35 | −0.76 | 0.65 | −0.61 | 0.51 |

Results of our statistical models 3a and 3b, testing for the effect of contexts on peering, including the dependent variables, the effects with associated estimated error and credible interval at 95 (in bold if they did not cross 0) and 89%.
[a]Group East.
[b]Extractive foraging without tools as reference categories, other contrasts are shown in (Table S3).

that, as observed in humans, chimpanzee social learning strategies based on the preferred age of role models changes through development[85]. Although the function of peering could change across the lifespan, immatures show an increased preference for older, non-mother role models. This preference, especially in non-extractive contexts, which encompasses rare and diverse food items[42], suggests that tolerant role models offer valuable learning opportunities that mothers rarely provide, which might facilitate diet expansion. Our results mirror several observations from other species, notably in a sociable and tool-using population of orangutans *Pongo abelii*[36]. Similarities with *Pongo abelii* include high frequencies of peering events directed at several role models in feeding contexts, with a peak around weaning age and an increase of peering with task complexity[26]. Resemblances in peering behaviours between *Pan troglodytes verus* and *Pongo abelii*, compared to *Pongo pygmaeus wurmbii*, which uses fewer tools and whose infants have limited social opportunities beyond the maternal unit, support the idea that social tolerance and access to diverse role models may facilitate the acquisition of complex skills. Across chimpanzee populations, the size and complexity of toolkits vary, as does tolerance[50,91]. Whether these factors are linked remains to be tested.

In conclusion, our study supports the idea that peering is used by chimpanzees when observational social learning is expected, especially when individual learning of skills is difficult. Wild chimpanzee social organisation allows immatures to peer at their mothers and at a diverse range of role models, at least through the first decade of life. Optimal exploitation of these opportunities requires efficient and flexible socialisation and learning strategies, and our results suggest that chimpanzees adjust their role model choices depending on the feeding context and developmental stage. High maternal tolerance might be key for tool use acquisition while availability of additional tolerant role models may further facilitate acquisition of the highly diverse chimpanzee diet, the range of tool using techniques, and potentially tool use proficiency. Some food items are rare or highly monopolizable, and mothers might not always be the individual foraging or the possessor of such foods. Thus, opportunities to observe rare feeding techniques increase when, in addition to mothers, non-kin role models allow immatures to peer at them while feeding. Our results fit with the idea that tolerance of the social environment may facilitate the emergence of both complex and expansive toolkits, which are common in some chimpanzee populations. A further test of this idea would be to determine if more tolerant or socially diverse chimpanzee populations have larger, more complex toolkits. Additionally, whether the benefits gained by immatures peering at their mothers in tool use contexts throughout development contributes to the evolution of a prolonged dependency on mothers during the first decade of life remains to be tested. Taken together, social tolerance of mothers and

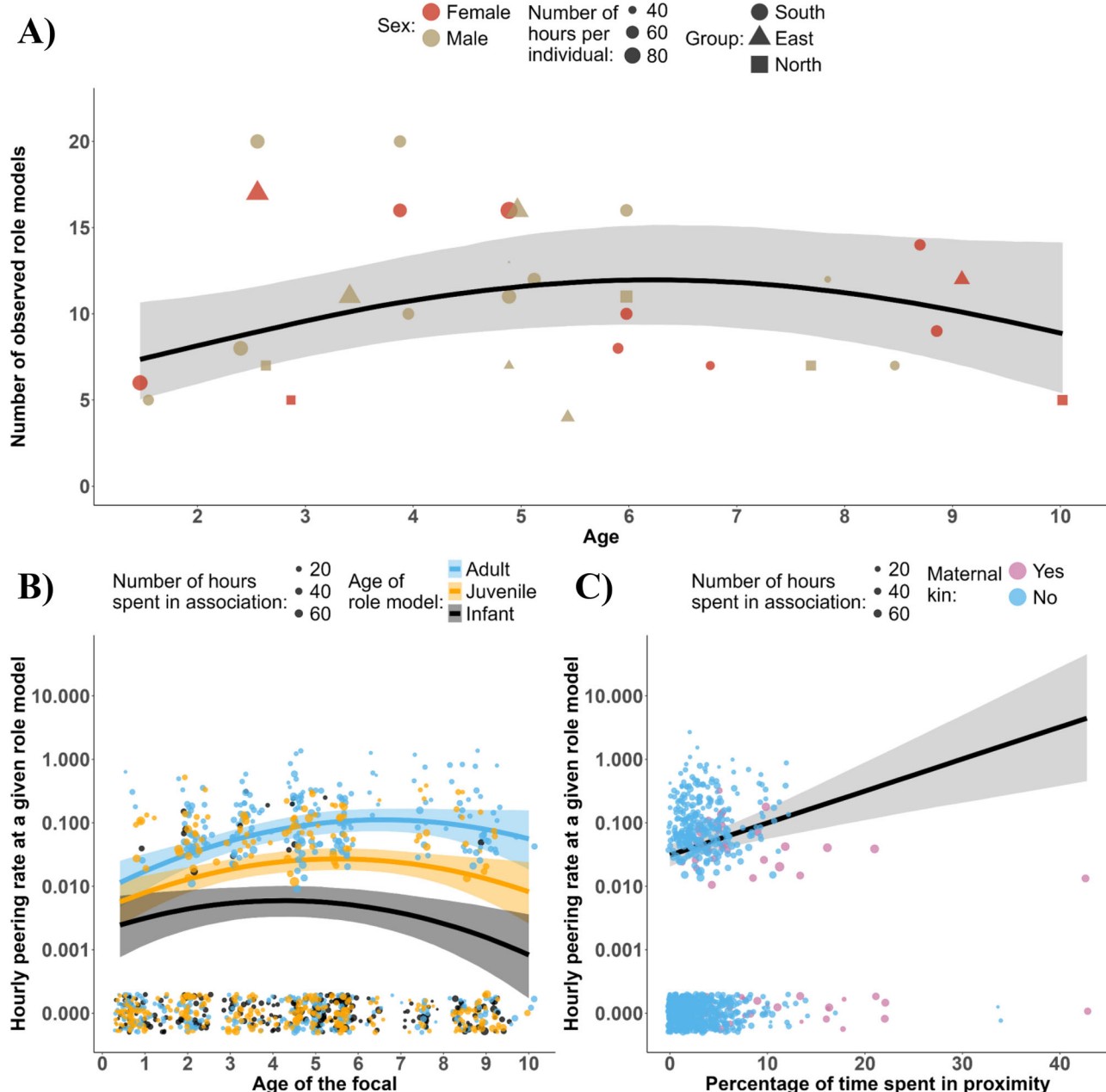

**Fig. 4 | Effect of development on the number and the characteristics of role models peered at (age and social tolerance). A** Effect of age on the number of non-mother role models peered at (model 4a), predicted for 60 h of observation. Each dot represents the number of role models peered at per subject. **B)** Effect of the interaction between focal's age and receiver's age on the probability of peering at a specific role model (model 4b). Infants include all role models less than 4 years, juveniles include role models between 5 and 12 years, adults include all role models more than 12 years. The regression lines are based on the mean age for each age class, respectively 1.6 years, 7.3 years, and 26 years. **C)** Effect of the percentage of time spent in less than 1 m proximity, as a proxy of social tolerance, on the probability to peer at a specific role model (model 4b). The regression lines from the Bayesian Regression model show a 95% credible interval. The dots in **A** represent the number of observed role models per individual per observation time. The dots in **B**, **C** represent the frequency of peering events directed at a specific individual per association time within a dyad; a jitter function was used to reduce overplotting.

others could be key factors allowing animals to diversify their feeding strategies and to acquire flexible tool use skills.

## Methods
### Data collection
The study focused on three well-habituated chimpanzee communities (North, South, and East) at the Taï Chimpanzee Project, Côte d'Ivoire[92], between January 2021 and April 2023, with a mean observation period between the first and the last day of focal of 10.10 (range 0–14) months per individual. We conducted half-day focal follows[93] (mean = 3.28 + /−2.24 h per individual a day) for all non-orphan male and female immatures (below the age of 10 years), totalling 35 individuals with known ages (see Table S1). Over the study period, we included five new-borns in the data, and although no immatures died, we stopped collecting data on two individuals due to the disappearance of their mother. We chose the order of the focal pseudo-randomly, with priority given to less observed individuals to balance observation hours across subjects. We reported continuously the activity and the social interactions of the focal individual and documented all changes in party composition and

**Table 4 | Models testing developmental effects on the number and the characteristics of peered role models (age and social tolerance)**

| Prediction | Response | Term | Estimate | SE | 95% CI | 95% CI | 89% CI | 89% CI |
|---|---|---|---|---|---|---|---|---|
| Model 4a) Number of observed role models will peak around weaning | Number of observed non-mother role models | Intercept | −1.66 | 0.12 | −1.89 | −1.43 | −1.85 | −1.48 |
| | | **Age** | **0.15** | **0.07** | **0.00** | **0.29** | **0.03** | **0.27** |
| | | **Age$^2$** | **−0.14** | **0.07** | **−0.27** | **0.00** | **−0.25** | **−0.03** |
| | | Group North[a] | −0.22 | 0.20 | −0.63 | 0.18 | −0.56 | 0.11 |
| | | Group South[a] | 0.07 | 0.13 | −0.18 | 0.33 | −0.13 | 0.28 |
| Model 4b) Peerers seek tolerant role models older than themselves | Number of peering events directed at non−mother role models | Intercept | −5.70 | 0.60 | −6.90 | −4.53 | −6.67 | −4.76 |
| | | Age peerer | −0.24 | 0.44 | −1.10 | 0.62 | −0.93 | 0.45 |
| | | **Age receiver** | **0.95** | **0.10** | **0.76** | **1.16** | **0.79** | **1.12** |
| | | Maternal kin (No) | −0.20 | 0.36 | −0.90 | 0.52 | −0.77 | 0.38 |
| | | Dyadic association index | −0.12 | 0.13 | −0.38 | 0.13 | −0.33 | 0.08 |
| | | **Age peerer$^2$** | **−0.42** | **0.15** | **−0.73** | **−0.12** | **−0.67** | **−0.18** |
| | | Group North[a] | −0.03 | 0.43 | −0.88 | 0.82 | −0.70 | 0.67 |
| | | Group South[a] | 0.37 | 0.31 | −0.25 | 0.99 | −0.12 | 0.87 |
| | | Sex receiver (Male) | −0.07 | 0.25 | −0.56 | 0.41 | −0.46 | 0.32 |
| | | Sex peerer (Male) | 0.11 | 0.28 | −0.46 | 0.66 | −0.35 | 0.57 |
| | | **Proportion of time in less than 1 m** | **0.44** | **0.11** | **0.21** | **0.65** | **0.26** | **0.61** |
| | | **Age peerer:Age receiver** | **0.26** | **0.11** | **0.07** | **0.47** | **0.10** | **0.43** |
| | | Age peerer:Maternal kin (Yes) | 0.13 | 0.35 | −0.55 | 0.82 | −0.43 | 0.67 |
| | | Age peerer:Dyadic association index | −0.05 | 0.11 | −0.28 | 0.17 | −0.23 | 0.13 |
| | | Sex receiver:Sex peerer | 0.02 | 0.28 | −0.53 | 0.57 | −0.42 | 0.44 |

Results of our statistical models 4a and 4b, testing for the effect of age on the number of observed role models, including the dependent variables, the effects with associated estimated error and credible interval at 95 (in bold if they did not cross 0) and 89%.
[a]Group East as reference category, other contrasts are shown in (Table S3).

size. Party size was defined as all the individuals present within 30 meters of the focal individual[94]. Additionally, we recorded the activity of the mother and the proximity of the focal to all other individuals as 5 min scans. We collected this data using the CyberTracker software (v.3.527). Data was collected by four observers with inter-observer reliability tests indicating good accuracy between them for focal activity (Cohen's Kappa tests: $\kappa = 0.72; 0.81; 0.70$) and party-composition (Cohen's Kappa tests: $\kappa = 0.89; 0.80; 0.78$)[95].

Maternal rank, dyadic association index, and proportion of time spent in close proximity were obtained using both long-term data and data collected for this study. Each researcher collecting the long-term data at the Taï Chimpanzee Project is trained by the project's longest-serving field assistant (H.N.K.), the most experienced field assistant working in the community and the co-director of the project (L.S.). Before participating in long-term data collection, researchers pass an inter-rater reliability test with an experienced field assistant observing the same individual for at least 4 h, which assesses their ability to identify individuals and accurately record behaviours. Once a satisfactory level of agreement is reached, typically over 85% for individual recognition and behavioural data, the researcher can start to collect long-term data[92].

### Data set
Overall, we recorded a total of 1873 observation hours during 570 focal half-days. **Peering events** were defined as directly looking at the task engaged in by another individual at close range (<2 m) for >5 s)[26], facilitating clear observations of the task. Peering was directed towards both feeding behaviours and social behaviours (Fig. 1A). However, our analyses focused solely on peering events occurring in feeding contexts given that feeding behaviours were the predominant targets of peering. Note that although some chimpanzee social behaviours are known to be influenced by social

learning[96,97], their dynamic nature likely makes them less amenable to peering.

Explicit **begging solicitations** included "extended hand" begging gestures, whimpers, and direct contact with the target or its items while the target was processing food. Whilst peering may partially function to solicit food, it is a more passive social interaction than our categories of food solicitations. Therefore, by differentiating between peering and explicit begging we could explore how these different social interactions around food items (i.e., peering only, explicit begging only, and both) vary in relation to the food type and the identity of partners (see prediction 1). For each peering and explicit begging event, the identity of partners, the solicited food item, and the outcome of the solicitation event (i.e., if the solicitor obtained the food or not) were noted. Consecutive peering events or explicit begging events were scored as two distinct events if the focal individual changed his activity or if its target started the processing of a new item in between them. Examples can be seen in the following video: https://youtu.be/FBOpYsf7shg.

We defined several types of processing for each type of food item, as shown in Table 5.

**Extractive foraging attempts** were defined as the manipulation of an embedded food item. The outcome of the foraging attempt was considered successful if the individual managed to extract edible parts and unsuccessful if the individual used different items or engaged in another activity without having been able to extract food.

### Statistics and reproducibility
Data preparation and statistical analyses were conducted in R 4.3.0 using the RStudio Interface[98]. To test our predictions, we performed a series of Bayesian regression models and survival analysis in R using the "brms" package[99]. In all the models, we z-transformed continuous predictors to a

**Table 5 | Definition of food processing complexity and food monopolizability from the perspective of adult chimpanzees**

| Type of processing: | Definition: | Example: |
|---|---|---|
| Low monopolizability | Multiple spread-out food clusters, such that only a single cluster is defendable by one individual. | Leaves, fruits, seeds. |
| High monopolizability | Food clumped in one piece or cluster, defendable by one or few individuals. | Meat, honeycomb, *Treculia africana*. |
| Non-extractive | Non-embedded food items. | Leaves, fruits with soft skin. |
| Extractive without tools | Food items embedded in hard non-edible matrices that do not require tools to be extracted. | Eggs of ants, termites, seeds within pods. |
| Extractive with tools | Embedded food items that require tools to be extracted. | Larvae extraction, nut cracking. |

mean of zero and a standard deviation of one to facilitate convergence and interpretation of the results. All offset terms that were included in models with "Poisson" and "Negative binomial" distributions were log transformed. Collinearity assessment was performed using the "vif" function from the "car" package[100]. To reduce type 1 error, maximal random slope structures were added in all the models[101,102].

For every model, we ran 3000 iterations with a "warm-up" of 1500 iterations over 8 Markov Chain Monte Carlo (MCMC), leading to 12,000 posterior samples[99]. We fitted models using weakly regularizing priors, specifying normal (0,1) distributions for the fixed effects and Student's t-distributed (3, 0, 2.5) priors for the random effects. Additionally, we applied uniform LKJ(1) priors to the covariance matrices of the random slopes. We observed no divergent transitions after warm-up, and model convergence and appropriate mixing of chains were indicated by trace plots. Rhat values inspection of all MCMC revealed satisfactory values (<1.01)[103]. Posterior predictive checks were executed using the "pp_check" function of the "brms" package see Fig S2. For all models, we report the estimate (mean of the posterior distribution) and credible intervals (CI) at: CI89% and CI95% allowing us to discuss strongly supported and "weakly" supported effects altogether without having to take a dichotomous approach (effect or no effect)[104]. Result plots were generated using the package "ggplot2"[105] and the conditional effects function of "brms".

For the survival analyses, we used the package "fitdistrplus"[106] to compare distributions of the non-censored data with theoretical distributions with a lognormal structure, which indicated a good fit for both models (supplementary material).

## Models

The list of all the models is organised according to the four questions posed in the introduction, with corresponding prediction, dependent variable, fixed effects, random factors, and family distribution provided in Table S2.

In model 1a, we predicted that peering will elicit less food transfers than explicit begging. To test the effect of the type of social interaction in feeding contexts (i.e., peering, explicit begging, or both) on the probability of receiving a food transfer (yes/no), we used a Bayesian Generalized Linear Mixed Model (GLMM) with a Bernoulli distribution. We removed from the dataset 107 out of 3001 social interactions where we could not establish if food transfer occurred or not. We included the age and the age squared of the focal as fixed effects as we expect transfers to peak around weaning age[69]. As mothers and other kin but less so might be more likely to share food than other individuals[68], we included the kinship identity of the partner (mother, maternal kin, other) as a fixed effect. As the number of processing steps required to access the food is known to influence maternal food transfers in explicit begging context in orangutans[69], we also included the type of foraging as a fixed effect in the analysis in interaction with the type of social interaction. Our model included a total of 3001 events of 35 individuals.

In model 1b, we predicted that peering in tool use contexts will induce a higher probability of manipulating tools than explicit begging. To test the effect of social interaction type in tool use context on the subsequent probability of manipulating tools, we used a survival analysis with a lognormal family distribution. We only included social interactions occurring in tool use contexts to facilitate the distinction between food processing and direct feeding attempts. We controlled for the offspring's age (as linear and squared terms). As mothers, but not other individuals, might facilitate access

to tools[46], while the number of individuals present could increase feeding competition, we included the kinship identity of the receiver (mother/non-mother) and the party size as control factors. Our model included a total of 856 events of 31 individuals.

In model 1c, we predicted that foraging complexity will increase the relative use of peering compared to explicit begging. Immatures mainly interact in feeding contexts to access resources they cannot obtain alone. This may be because the resource is rare or monopolized or because they lack the skills to process it. In the latter case, gaining information on processing is beneficial. Thus, if peering provides information, it should increase in proportion with task complexity. To test the effect of processing complexity (i.e., non-extractive, extractive, extractive with tools), on the type of social interaction (i.e., peering, explicit begging, or both) we used a multinomial model with a categorical distribution. We included the age and age squared, as weaning might have distinct effects on informational and nutritional needs, and the sex of the offspring and the kinship identity of the partner (towards mother/non-mother) as fixed effects. Our model included a total of 3102 events of 35 individuals.

In model 2a, we predicted that the frequency of peering events in foraging contexts will peak around weaning age, when most of the foraging skills are being acquired, and continue later in development. To test the effect of age on peering rate (i.e., peering events per hour of focal), we used a GLMM with a negative binomial distribution as the dependent variable was over dispersed. As we expected a peering peak in development, we included an age squared term. As sex might influence weaning age and learning needs, we included a two-way interaction between sex and age and age squared. We included an offset term (log transformed) to account for the length of the observation period. Our model included 570 half-day focals of 35 individuals.

In model 2b, we predicted that food items that are difficult to process or access will elicit a higher frequency of peering events. To test the effect of food processing complexity (i.e., non-extractive, extractive, extractive with tools, Table 1) and of resource monopolizability (high/low, Table 1) on peering rate (i.e., peering events per hour of focal) when at least one adult (the mother) is feeding, we used a GLMM with a Poisson distribution. To identify situations in which individuals had the opportunity to peer (events), the activity and the type of food consumed by the mother were recorded via scan every 5 min. The event was stopped if the mother changed her activity or if the focal travelled for more than 60 s. Peering events directed at individuals feeding on different types of food items during the event were removed from the analysis (31 out of 1414). We included an offset term (log transformed) to account for the time spent in the food patch. Our model included 2697 foraging events of 35 individuals.

In model 2c, we predicted that failures in one's feeding attempts will subsequently reduce the latency before peering. To test whether food extraction failure of immatures decreased the latency between the outcome of their extraction attempt and their next peering event towards an individual feeding on a similar item we used a survival analysis with a lognormal family distribution. We restricted observations to two specific situations where food processing was easily observable and that occurred numerous times, specifically pod opening and nut cracking. We included the type of task (i.e., nut cracking, pod opening) as a control factor. Food processing attempts on pre-processed items or with uncertain outcomes (133 out of 1413) were excluded from the analysis. Data were censored if the offspring

moved out of the foraging patch or engaged in other foraging attempts before a peering event occurred. Our analysis included 1280 events of extractive foraging attempts by 30 individuals, as five of our subjects were not observed engaging in extractive foraging.

In model 3a, we predicted that the proportion of peering directed at the mother will decrease with age as immatures acquire their own skills and other models become preferred. To test the effect of age on the probability to peer at other individuals rather than at the mother, we used a GLMM with a Bernoulli distribution. We integrated a two-way interaction between age and party size because if non-mother role models become preferred with age, party size should have a larger effect on the probability of looking at other models than the mother. Age squared was not included in this model because we expected the proportion of peering directed at the mother to decrease gradually as offspring become independent. We selected peering events occurring while the mother and at least one other weaned individual (>4 years) were present in the party composition. This included a total of 2024 peering events of 35 individuals.

In model 3b, we predicted that immatures will preferentially look at their mothers over alternative role models for the acquisition of complex feeding skills. To ensure that most immatures had the opportunity to look at other individuals while the mother was feeding, we looked at the probability of having at least one individual feeding at the same time using a subset of our data (681 events) for which we recorded feeding behaviour also of non-mother individuals. The probability of having other individuals feeding at the same time when at least one other weaned individual (>4 years) was present in the party composition was 90%, suggesting that immatures often had opportunities to look at alternative role models. To test the effect of processing complexity (i.e., non-extractive, extractive, extractive with tools) on the probability of peering at the mother (yes) rather than at other individuals (no), we used a GLMM with a Bernoulli distribution. To control for role model choice opportunities, we only included events during which mothers and other individuals were present, and at least one individual (the mother) was feeding. Given that the probability of having suitable role models should increase with party size, we included a two-way interaction between task complexity and party size to test if biases toward the mother persist when party size increases. We also integrated a two-way interaction between the processing complexity and the age of the focal as we expected that processing complexity could bias the attention toward the mother and, therefore, reduce the effect of age on the probability of peering at non-mother role models. We selected peering events occurring while the mother and at least one other weaned individual are present in the party composition. Our model included 1304 peering events of 35 individuals.

In model 4a, we predicted that the number of observed role models will peak during development. To test the effect of focal's age on the number of non-mother models they observed (i.e., peered at) in feeding contexts, we used a GLMM with a Poisson distribution. We examined this question by aggregating focal observation time across the entire period of the data collection (mean = 60 h), ensuring ample opportunities to observe all group members. Because observation time might have a non-linear effect on the number of observed group members, we removed from the analysis five individuals with a small sample size (i.e., observed less than 11 h). As we expected a peak around weaning age, we used the median age and age squared of the focal individuals. Our model included 30 individuals and used observation time as an offset term (log transformed).

In model 4b, we predicted that peerers will preferentially seek tolerant role models older than themselves. To test the effect of social tolerance within a dyad and of partner's age on the number of peering events the focals directed at different non-mother partners, we used a GLMM with a negative binomial distribution as the dependent variable was overdispersed. We log-transformed the age of the receiver variable because the impact of age on peering is not expected to be consistent across different age ranges. We used metrics of close proximity (proportion of observation time spent at less than 1 m) as a proxy of social tolerance between partners. For each dyad, the proximity score included the number of scans spent in less than 1 m proximity, divided by the number of scans in which both individuals were

observed in the party composition. Proximity was recorded in all contexts and combined both data collected for this study and long-term project data collected during the same period, reducing, therefore, the risk of having scans occurring in the peering contexts used in this study. For each dyad, the percentage of scans for which the receiver was in close proximity and received a peering consecutively was lower than 1%. To reduce estimation biases of the time spent in proximity, we excluded dyads that appeared in less than 10 scans. To disentangle the effect of spatial proximity from a possible effect of familiarity, we also included for each dyad the dyadic association index obtained using the simple ratio index on the observation period[107] (Time A and B spent in the same party)/(Time A seen in parties + Time B seen in parties − Time A and B spent in the same party). To ensure that most potential role models were available sufficiently to provide peering opportunities, we aggregated observation time by dyad over the period of the data collection, leading to a mean of 16.6 h per dyad, and we controlled for it using an offset term. For each dyad, we used the median age and median age squared of the focal and the median age of the partners as fixed effects. We also included a two-way interaction between the age of the focal and the age of the partner, as focal might benefit from observing older individuals than themselves. We also included a two-way interaction between the sex of the focal and the sex of the partner as attentional biases toward same sex individuals have been reported in orangutans[108]. As maternal kin might be more accessible than other role models[109], we included maternal kinship as a control factor (binary variable: yes = is a maternal sibling, nephew, uncle, grandmother; no = other). We also included a two-way interaction between focal's age and maternal kin as older immatures might exhibit a bias toward less familiar role models. Our model included 1082 dyads. We also integrated the identity of the partner as a random intercept.

In addition to the control factors described in each model, several variables were added. As we expected that maternal dominance could have long and short-term effects on the access to food resources and to other role models, we included maternal rank as a control variable in models 1a, 1c, 2a–c, 3a and 3b, but not in model 4a and 4b for which mothers were excluded, or for model 1b for which access to resources was already controlled for. We estimated maternal rank based on the dominance hierarchies calculated using long-term data on pant-grunts and applying a modification of the Elo-rating method[110] developed by ref. 111 see details in ref. 94. We also included the party size per event or the average party size per day to control for role model availability and food competition in models 2a–c, 3a, 3b and S1b, but not in other models as data were either aggregated over a long period of time (4a and 4b), or because only the type of social interaction or the response to solicitation were tested (1a and 1c). In models 3a and 3b, in which we tested the probability of peering at the mother compared to other role models, we added the presence of maternal kin in the party as they could provide tolerant alternative role models and restrict the access to the mother as in the case of younger siblings. To account for possible variation across groups, we added group identity ("East", "North", or "South" group) as a fixed effect in all the models. To account for possible sex biases in the use of peering behaviour, we included the sex of the subject as a fixed effect in all the models. In model 2a, we integrated a two-way interaction between the age and sex of the focal as we expected that the use of peering behaviour during development would be influenced by the sex of the immature. As we found no consistent effect of this interaction in model 2a, we removed it in models 2b and 2c. In model 3a, we included a two-way interaction between the age and party size of the focal as we expected the influence of the party size to be positively correlated with the age of the individual. As we found no consistent effect of this interaction in model 3a, we removed it in model 3b.

In models 1a–c, 2a–c, 3a, 3b, and 4b, we included a random effect intercept for the identity of the focal. We also included an intercept for the identity of the mother since 12 out of 21 mothers had multiple offspring in the dataset in models 1a–c, 2a–c, 3a, 3b, but not in 4a and 4b, as mother-offspring dyads were excluded from the models. For models 1a–c, 2b, 2c, 3a, and 3b that included multiple events from the same observation day, we included the date of the observation.

## Data availability
The data used for the analysis of the current study are available on Zenodo at: https://doi.org/10.5281/zenodo.13951539[112].

## Code availability
The codes used during the current study are available on Zenodo at: https://doi.org/10.5281/zenodo.13951539[112].

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

## Acknowledgements

We thank the Ministère de l'Enseignement Supérieur et de la Recherche Scientifique and the Ministère de Eaux et Fôrets in Côte d'Ivoire, and the Office Ivoirien des Parcs et Réserves for permitting the study (Wittig/006/MESRS/DGRI). We are grateful to the Centre Suisse de Recherches Scientifiques en Côte d'Ivoire and the staff members of the Taï Chimpanzee Project for their support. The Taï Chimpanzee Project received core funding from Max Planck Society provides core funding for the since 1997. We are extremely grateful to Kim Vermeulen and Giuliana Centofanti, two research assistants who contributed to the data collection for one year. We thank the local field assistants for collecting long-term data and all the staff members of the Taï Chimpanzee Projects for their help in the field. We are indebted to Christophe Boesch's efforts in establishing and nurturing the Taï Chimpanzee Project and tremendously advancing western chimpanzee conservation. This study was funded by the Hominoid Brain Connectomics Project through the Max Planck Society (M.IF.NEPF8103 and M.IF.E-VAN8103) and the European Research Council (ERC) under the European Union's Horizon 2020 research and innovation program awarded to C.C. (grant agreement no. 679787). This study was part of O.N.L. PhD project funded by Université Claude Bernard Lyon1.

## Author contributions

The authors confirm contribution to the paper as follows: study conceptualization: O.N.L., C.C., R.M.W.; data curation: O.N.L., E.R.; formal analysis: O.N.L; funding acquisition: C.C., R.M.W.; methodology: O.N.L., C.G.B., C.C., R.M.W., L.S.; supervision: C.C., R.M.W., P.F.F.; writing original draft preparation: O.N.L.; writing review & editing: O.N.L., L.S., C.G.B., P.F.F., C.C., R.M.W., E.R. All authors reviewed the results and approved the final version of the manuscript.

## Funding

## Competing interests

The authors declare no competing interests.

## Ethical approval

We have complied with all relevant ethical regulations for animal use. This study is purely observational and non-invasive, and has been

approved by the Ethikrat der Max Planck Gesellschaft (04/08/2014). At the Taï Chimpanzee Project we follow the IUCN best practice guidelines for health monitoring and disease control in great apes, including rigid hygiene rules, wearing a surgical mask while following the chimpanzees and keeping 7 m distance. In addition, we employ a 5 day quarantine before observing the chimpanzees to protect their health.
