## [Transparent Peer Review file · Communications Biology]

Social tolerance and role model diversity increase tool use learning opportunities across chimpanzee ontogeny.

Corresponding Author: Dr Oscar Nodé-Langlois

Version 0:

Reviewer comments:

Reviewer #1

(Remarks to the Author)

This paper documents the developmental trajectory of peering behaviors in a wild population of chimpanzees. Peering (i.e., attentively watching a conspecific engage in feeding or other noteworthy behaviors) peaked around weaning age, when mastering feeding behaviors is presumably critical, and was more common when engaging in complex feeding tasks. Peering was primarily directed towards mothers during early development, but immature chimpanzees of all ages relied on their mothers most when observing more complex feeding behaviors, such as tool use. Immatures' social milieu expanded as they approached adulthood, as older immatures were less reliant on their mothers for gaining this social information. The manuscript's theoretical grounding is sound, the dataset is impressive, and the associated analyses are thorough and rigorous. The paper would make an excellent contribution and is highly relevant to all who are interested in how complex skills are socially learned in human and non-human animals. My minor comments below mainly focus on areas where the clarity of the manuscript could be somewhat improved.

MINOR COMMENTS

L12: "with development" might read more clearly as "across development"

L27: "for example when" can be replaced with "because" for ease.

L32-34: This could be more "list-like" to guide the reader, as it took me a couple reads to count up the factors. Also, it might be helpful to list these factors in order that matches that of the subsequent paragraphs (i.e., protracted development first, then socially tolerant carers, etc.).

L52-62: I'm not sure how useful this paragraph is. In its current form, it breaks up the natural flow set up by the list of factors laid out in L32-34, and I'm not sure what main takeaways I should be drawing from it. If some of this information is crucial, it might fit more naturally in the paragraph focused on the importance of having tolerant role models (L75-89).

L89: This is pedantic, but by 'older', do you simply mean adult? Or older at the population-level?

L94-97: The division between the list of foods and the list of materials is somewhat awkward. Consider breaking this up into two sentences.

L150-151: When in development do you expect this to peak? Mid-development around the predicted weaning peak? A more specific prediction would be helpful here.

L156-158: When specifically were these data collected? It would be good to know just how much these immatures aged over the course of the study. Also, were any individuals dropped or right-censored due to death?

L180-182: This is oddly worded. Maybe note that peering was directed towards both feeding behaviors and social behaviors. However, given that (1) feeding behaviors were the predominant targets of peering and (2) mastering these complex tasks may be more reliant on social learning, the analyses focused solely on these peering events.

L244-246: How many times did this occur? I presume that this is a very small portion of the dataset?

L257-258: Does this mean that 5 individuals in the sample were always successful at extracting food? Or were other exclusion criteria more important here?

L261-263: Why was this calculated with a small portion of the total peering events? If I'm following correctly, shouldn't this be easily calculable from the complete sample of peering events?

L287: Why is this sample so different from that described in L275? The criteria otherwise seem identical, unless I'm missing something.

L289: Same comment regarding the prediction as above.

L296: If two individuals were removed, shouldn't the sample be 33?

L312: While dyadic association indices are commonly used in chimpanzee research and for other fission-fusion species, it might be helpful to quickly define it here.

L321: How was kinship measured? As a binary category?

L323-324: Were focal peerers within the dataset ever the targets of peering? I.e., are the two random effects independent pools of individuals?

L336: Were these included as fixed effects (and not random effects) because the groups were too few to estimate these variances?

L378-379: Why is this effect conditional on 15 minutes having passed? In other words, if the effect size is constant (as in a proportional hazards model), wouldn't the predicted probability for failures be double that of successes regardless of how much time has passed?

Figure 2: Why are the y-axes of panels B) and C) so different? Because the points are aggregated using different criteria?

L436-438: "per 50 h" vs. "per 60h" makes it hard to compare these predicted values.

Figure 4: The figure itself skips from A) to C). Also, the y-axis scales on panels C) and D) are a bit hard to read, especially since the 0.001 marker is truncated to 0.00. Would it be possible to keep the maximum y-values the same across the two panels? Or are the points differentially aggregated?

L534-535: This sentence reads as somewhat repetitive with the concluding sentence in L530-533. Consider editing to avoid redundancy.

L540-542: Passive voice is odd here.

L591-592: It seems more intuitive to me that this analysis could be recapitulated within populations (i.e., that immatures who are more widely or regularly tolerated will grow a wider toolkit of complex behaviors than those that lack as many opportunities to peer at unrelated adults). Is there stark variation in social tolerance across chimpanzee populations?

Reviewer #2

(Remarks to the Author)

This paper describes how patterns of peering behaviour in immature wild chimpanzees are associated with age, role model and feeding complexity. Overall the results are clearly presented and described and the paper makes a novel contribution to the literature in this area. The figures are clear and helpful in graphically representing some of the key findings. My comments relating to this paper are below.

Title and abstract

It would be helpful in either the title or abstract to provide the scientific name for this chimpanzee species given patterns of tool use vary between sub-species of chimpanzee.

It would be helpful to provide the age range and sex of the immature chimpanzees in the abstract.

Introduction

The introduction does a good job of reviewing the relevant literature and explaining the importance of peering behaviour in the context of social learning.

Lines 109 – 130 – It would be helpful in this paragraph to explain more explicitly the novelty of the current study as compared to previous studies on peering behaviour in immature wild chimpanzees. Is this the first study to examine this behaviour across the whole range of food types?

Line 133 – It would be helpful to provide the weaning age here to give more specificity to the prediction

Line 140 – It is not really clear how this prediction relates to ‘tolerant’ mothers. The tolerance of the mothers towards the infants is not measured or mentioned in either of the predictions.

Methods

Data collection

It would be good to provide some more specific information on the 35 individuals here or in the SI, including how many hours of observation per individual, over what time period were individuals followed, the mean age of each individual across the samples for that individual and the number of each sex.

It is not clear from this description the time period over which this data was collected i.e. were all observations of a specific individual done when they were the same age, or were observations across multiple years?

Line 168 It would be good to explain how the dyadic association index was calculated here given there are different ways of calculating this.

Data set

Line 179 – When describing the peering events, it would be helpful to explain that these were calculated as peering events per hour of the focal. This is explained in line 230 but may be clearer here. Were all peering events treated the same regardless of duration and was the duration recorded? Presumably peering events could vary in length from >5s to over a minute?

Statistical analysis

As some readers may be unfamiliar with Bayesian regression models, it may be good to explain why these were chosen over frequentist approaches for this study. Similarly, for readers unfamiliar with a Bayesian approach, it may be helpful to give a brief explanation of how to interpret the 95% and 89% credible intervals, as these differ from the confidence intervals which readers may be more familiar with

Line 212 – give the full name for MCMC chain on first usage

Models

Line 230 – give full name of glmm on first usage

Line 326 Control fixed effects

The model numbering here does not align with the paper. For example, on line 329 there is a model 2c but this does not appear in the previous section. Further, there are models 4a and 4b which presumably relate to the models presented in the SI.

There needs to be a more explicit justification for why some control variables were included in some of the models but not others. I'm sure there are good reasons for this based on the other variables included in the models, but it would be good to set these reasons out for the reader.

Results

Results tables

It may be clearer to use a decimal point rather than a comma in the Results tables.

It could be made clearer in the Table title and/or text what the bold and italics signify i.e. presumably that the credible interval does not cross zero?

It could also be made clearer in the Tables (if this is correct) that the response variable is not the ‘raw’ number of peering events, but the peering rate per hour of focal observation

Line 372 – I may be reading Table 2 or the text incorrectly, but the estimates for extractive foraging in the text do not appear to be consistent with Table 2. In the Table 2, for Model 1b, extractive with tools has an estimate of 0.61, but in the text the estimate for extractive with tools is 1.15?

Two of the headings in the results section (Early learning and learning of complex tasks rely on tolerant mothers; later learning relies on many tolerant role models) seem more like an interpretation of the function of peering behaviour rather than a summary of the actual results reported. It may be better to have headings for these sections that relate more closely to the response variables in the models (peering behaviour), as learning of food processing skills (e.g. increase in processing efficiency) was not directly assessed in these models

Model 2a, Figure 3A

Is this reduction in proportion of peering directed at the mother a function of reduced proximity to mother and increased proximity to other individuals as the immature chimpanzee ages? So it is not that the immature chimpanzees are actively 'seeking out' opportunities to learn from others, but they spend more time in closer proximity to others and therefore have more opportunity to display peering behaviour.

For example, if the proportion of time the immature chimpanzee spent in close proximity to the mother vs. other individuals was plotted against age, there may be a similar pattern with older individuals spending less time in closer proximity to the mother.

Model 2b

The key prediction for this model as set out in lines 144-145 and Table S2 is that 'Mothers will be more likely to be observed for complex tasks' (quote from Table 2a).

I may be mistaken, but the key test of this prediction would then seem to be the estimates for the variables 'Non extractive' and 'extractive with tools' in Model 2b i.e. is the identity of the receiver of the peering related to the complexity of the food processing. The credible intervals for both of these variables cross zero.

It should be made clear in the text whether the key prediction as set out in lines 144-145 and Table S2 is supported by model 2b or not. This also applies to the statements relating the task complexity and reliance on mother in the discussion.

Instead in the text (lines 414 – 417) and Figure 3b, the reported results relate to the interaction between role food processing complexity and party size. Whilst this is interesting, overall model 2b does not seem to support the core prediction that 'Mothers will be more likely to be observed for complex tasks' as in Table S2 and lines 144-145. This prediction related to the 'main effect' of task complexity, not the interaction between task complexity and party size.

There is also some inconsistency in the numbering of the models (2b or 3b) in this section in the text and table

Table 4 - Dyadic association index is listed twice in Model 3b with different estimates.

Fig 4 C and D – should second value on Y axis be 0.001?

Discussion

Line 472 – is the term caregiver here used instead of mother? As mother was used throughout the manuscript, it may be clearer to use this term as well here or explain what is meant by the term caregiver (i.e. does this include caregivers other than the mother)

Line 529 – is this statement supported by the model results? See comments on model 2b above.

Line 584 – I am not sure whether the results support the statement relating to high maternal tolerance, as maternal tolerance was not directly measured in this study as far as I can tell.

Overall comments

The finding that peering appears to function to acquire information more than food from others is an important one and is referenced in the abstract and the first subheading in the Discussion. But the analysis for this is presented in the SI. Unless there are word limit or space constraints, I would suggest moving this analysis to the main paper as it is only three additional models. In this case the results would have to be explained more fully as for the results in the main paper.

As part of the review, I am asked to comment on 'the ability of a researcher to reproduce the work, given the level of detail provided'. The models are clearly described, but I could not see a statement about availability of data and analysis code. It would greatly aid a researcher to reproduce the work if the data and analysis code were provided. Given there are no issues around anonymity and confidentiality as there sometimes are with human data, I see no reason why the data and analysis code should not be provided in line with open science principles.

Version 1:

Reviewer comments:

Reviewer #1

(Remarks to the Author)

The authors have done a superb and thorough job responding to reviewer feedback. The flow of the Introduction is much improved, some details in the Methods have been clarified, and the figures are now easier to interpret. Overall, the manuscript is well-written, the analyses are robust, and the discussion is cogent and insightful. I have very few remaining minor comments, which mostly focus on the organization of the Introduction.

MINOR COMMENTS

L27: Remove the "the" to keep it parallel with 3).

L46-68: Maybe I'm misunderstanding, but I'm still not following the flow of the argument here and the introduction of social tolerance merged with the discussion of caregivers. Are there species that might lack toolkits because their caregivers (e.g., mothers) are so rejecting that offspring cannot observe their mothers' feeding behavior and strategies? More broadly, I think the Introduction would be more clear to me if the text and references in L47-56 were placed after the discussion of caregivers (i.e., L56-68) and paired with the paragraph discussing the wider pool of role models (L69-83).

L84: "on" should be "in"

L146-147: I'm curious about the logic behind this hypothesis. Might it also be possible to expect offspring to engage in more explicit begging if the task is difficult for them to accomplish on their own? Or is this unlikely since the tasks are impossible without a decent mastery of the tools themselves?

Figure 4b, 4c: Can the scales be the same for the adjacent figures?

Reviewer #2

(Remarks to the Author)

The authors have done a very good job in responding to the reviewer comments. They have fully addressed all the points I made in the review and made their data and code available.

We are pleased to resubmit our revised manuscript for consideration in *Communications Biology*. We appreciate the reviewers' thoughtful and constructive feedback, which has significantly improved the quality of our work. We have carefully addressed all the comments provided and incorporated our changes into the manuscript (text colored in red). Below, we provide a list of the revisions made in response to the reviewers' comments.

List of changes:

Reviewer 1:

This paper documents the developmental trajectory of peering behaviors in a wild population of chimpanzees. Peering (i.e., attentively watching a conspecific engage in feeding or other noteworthy behaviors) peaked around weaning age, when mastering feeding behaviors is presumably critical, and was more common when engaging in complex feeding tasks. Peering was primarily directed towards mothers during early development, but immature chimpanzees of all ages relied on their mothers most when observing more complex feeding behaviors, such as tool use. Immatures' social milieu expanded as they approached adulthood, as older immatures were less reliant on their mothers for gaining this social information. The manuscript's theoretical grounding is sound, the dataset is impressive, and the associated analyses are thorough and rigorous. The paper would make an excellent contribution and is highly relevant to all who are interested in how complex skills are socially learned in human and non-human animals. My minor comments below mainly focus on areas where the clarity of the manuscript could be somewhat improved.

Thank you very much for these positive comments.

L12: "with development" might read more clearly as "across development"

L27: "for example when" can be replaced with "because" for ease.

L32-34: This could be more "list-like" to guide the reader, as it took me a couple reads to count up the factors. Also, it might be helpful to list these factors in order that matches that of the subsequent paragraphs (i.e., protracted development first, then socially tolerant carers, etc.).

Thank you for these suggestions, we integrated all of these suggestions into the text.

L52-62: I'm not sure how useful this paragraph is. In its current form, it breaks up the natural flow set up by the list of factors laid out in L32-34, and I'm not sure what main takeaways I should be drawing from it. If some of this information is crucial, it might fit more naturally in the paragraph focused on the importance of having tolerant role models (L75-89).

We merged the two paragraphs together to make the connection with the prediction clearer. We wanted to explain our rationale of why peering at tolerant models is beneficial especially for complex tasks given that peering at mothers could be influenced by various factors in addition to social tolerance (attachment, physical proximity).

L89: This is pedantic, but by 'older', do you simply mean adult? Or older at the population-level?

We apologise if this was unclear - here we mean not only adults, but also immatures older than our focal. In our model the receivers' age was also considered a continuous log-transformed variable. We made this choice as we expect that competency improves gradually through development and into adulthood. We also integrated a two-way interaction with the age of the peerer as we expect them to observe models older than themselves. Further explanations are provided in the methods (line 375).

L94-97: The division between the list of foods and the list of materials is somewhat awkward. Consider breaking this up into two sentences.

Thank you for this suggestion. We now broke this sentence into two parts, one explaining food and one explaining tools (line 89)

L150-151: When in development do you expect this to peak? Mid-development around the predicted weaning peak? A more specific prediction would be helpful here.

We now added the precision “around weaning age” to this prediction (line 165). We expect it to peak around weaning age (3-5 years) because that’s when most of the skills are being acquired.

L156-158: When specifically, were these data collected? It would be good to know just how much these immatures aged over the course of the study. Also, were any individuals dropped or right-censored due to death?

We added the following precision in the text line 174 “The data were collected between January 2021 and April 2023, with a mean observation period of 10.10 (range 0-14) months per individual. - Over the study period, we included five new-borns in the data, and although no immatures died, we stopped collecting data on two individuals due to the disappearance of their mother. ” We also included a table summarizing mean age and observation period of the sampled individuals in the supplementary (Table S1).

L180-182: This is oddly worded. Maybe note that peering was directed towards both feeding behaviors and social behaviors. However, given that (1) feeding behaviors were the predominant targets of peering and (2) mastering these complex tasks may be more reliant on social learning, the analyses focused solely on these peering events.

We included a precision in the text, line 202. “However, our analyses focused solely on peering events occurring in feeding contexts given that feeding behaviors were the predominant targets of peering. Note that although some social behaviours are known to be influenced by social learning (**van Leeuwen and Hoppitt 2023; Malherbe in press**), their dynamic nature likely makes them less amenable to peering.”

L244-246: How many times did this occur? I presume that this is a very small portion of the dataset?

This included 31 out of 1414 peering. We now reported that number in the text line 302.

L257-258: Does this mean that 5 individuals in the sample were always successful at extracting food? Or were other exclusion criteria more important here?

Not exactly, this is because some individuals were never observed engaging in the tested extractive foraging tasks, usually because they were too young or because the items were not present during the observations. We added the precision “as five of our subjects were not observed engaging in extractive foraging” in the main text line 313.

L261-263: Why was this calculated with a small portion of the total peering events? If I’m following correctly, shouldn’t this be easily calculable from the complete sample of peering events?

Unfortunately, the presence of non-mother individuals feeding was not monitored in the beginning of the data collection, so we could test this only on a subset of the data. We now note this in the methods: “for which we recorded feeding behaviour also of non-mother individuals”, line 318.

L287: Why is this sample so different from that described in L275? The criteria otherwise seem identical, unless I’m missing something.

Thank you for raising this point. This is because, in the model presented in line 322, we tested the choice of role models in all the conditions during which the mother and at least one dependent

individual was in the party-composition. This included a lot of events during which other individuals but not the mother were observed feeding (27% if at least one other individual was in the party, now reported in the discussion line 686). In the model presented in line 332 however, we wanted to test for the effect of task complexity performed by the mother. We therefore had to ensure that at least the mother was feeding, which resulted in a smaller sample (The estimation, line 320, confirmed that these points were most of the time associated with opportunities to peer at other role models).

We include the precision: “To control for role model choice opportunities, we only included events during which mothers and other individuals were present and at least one individual (the mother) was feeding. Given that the probability of having suitable role models should increase with party-size, we included a two way interaction between task complexity and party-size to test if biases toward the mother persist when party size increases.”, line 337-339, to make this point clearer.

L289: Same comment regarding the prediction as above.

L296: If two individuals were removed, shouldn't the sample be 33?

Yes, that was a mistake in the text, we have corrected it line 351.

L312: While dyadic association indices are commonly used in chimpanzee research and for other fission-fusion species, it might be helpful to quickly define it here.

We added the corresponding calculus in the text line 371. “(Time A and B spent in the same party)/(Time A seen in parties 370 + Time B seen in parties - Time A and B spent in the same party)”

L321: How was kinship measured? As a binary category?

Kinship included maternal siblings and maternal grandmothers or granddaughter and nephews or uncles if the mothers stayed in the community and was coded as a binary measure (1 for kin individual, 0 for non-kin individual). We included this precision in the main text line 380-381.

L323-324: Were focal peerers within the dataset ever the targets of peering? I.e., are the two random effects independent pools of individuals?

Yes, for immatures some peerers were also targets of peering. The pool of peerers and receivers therefore partially overlap. Note that if A peer B and B peer A, this will result in two data points. As peering direction is very asymmetrical within dyads, we think this is not a problem as factors explaining individual variations in the probability to peer or to be peer are likely to differ. For instance, interindividual variations in competency and in ability to access to resources likely have opposite effects on peering behaviour depending on if the individual is the peerer or the receiver.

L336: Were these included as fixed effects (and not random effects) because the groups were too few to estimate these variances?

Yes, there were only three communities.

L378-379: Why is this effect conditional on 15 minutes having passed? In other words, if the effect size is constant (as in a proportional hazards model), wouldn't the predicted probability for failures be double that of successes regardless of how much time has passed?

This is because we used a log-normal distribution (which was fitting the data best). Note that depending on the time following the attempt, the probability to peer might first increase after the event and then decrease (either because of a reduction of resources or because of a motivation to feed or gain information). We added more precision on how our data fitted with several theoretical distributions for survival analysis (Gamma, Weibull, log-normal) in the supplementary.

We added more precision line 474: “After 10 minutes, the predicted probability of having peered was 50% for failures versus 20% for successes, increasing to 60% versus 30% at 20 minutes and 75% versus 45% at 40 minutes.”

Figure 2: Why are the y-axes of panels B) and C) so different? Because the points are aggregated using different criteria?

Yes, we added a precision in the figure legend “for the shown type of processing per subject”, line 491. In figure B), the points are aggregated by food monopolizability, which was associated with a lower sample size and therefore a higher variation between individuals (2 events for the highest dot, which corresponded both to nut cracking involving monopolizable tools). In the figure C) they are aggregated by foraging complexity.

L436-438: “per 50 h” vs. “per 60h” makes it hard to compare these predicted values.

Thank you for pointing this out, it was a mistake. The presented result is per 60 h.

Figure 4: The figure itself skips from A) to C). Also, the y-axis scales on panels C) and D) are a bit hard to read, especially since the 0.001 marker is truncated to 0.00. Would it be possible to keep the maximum y-values the same across the two panels? Or are the points differentially aggregated?

We changed the labels to make the 0.001 visible on figure 4B and 4C. One of the dyads (which are dependent twins) are very often in close proximity, which drives the curves to the left. So, it would be difficult to plot the entire curve while keeping the maximum y values and keeping a good visibility.

L534-535: This sentence reads as somewhat repetitive with the concluding sentence in L530-533. Consider editing to avoid redundancy.

Thank you for this comment, we removed a part of the corresponding sentence.

L540-542: Passive voice is odd here.

We changed the sentence to: “It is likewise unknown whether relying on tool-use learning from their mothers contributes to the reduced growth and loss of reproductive success that chimpanzees experience when orphaned before adulthood” (Line 638)

L591-592: It seems more intuitive to me that this analysis could be recapitulated within populations (i.e., that immatures who are more widely or regularly tolerated will grow a wider toolkit of complex behaviors than those that lack as many opportunities to peer at unrelated adults). Is there stark variation in social tolerance across chimpanzee populations?

This is a great point. We added a precision line 679 “Across chimpanzee populations, the size and complexity of tool kits vary, as does tolerance (Musgrave et al., 2019; Wilson et al., 2014). Whether these factors are linked remains to be tested.” There is a little variation in the tool kits of individuals within communities, although there are interindividual differences in the frequency and the efficiency of tool use behaviours (potentially impacted by the tolerance of mothers and other role models) (Estienne et al. 2019; Berdugo et al. in press; Lonsdorf 2006). However, some populations display strong differences in social tolerance. For instance, the Taï chimpanzees, whilst fairly consistent across communities, exhibit lower levels of infanticide or inter-group lethal aggression compared to other populations, and females are generally more gregarious. Additionally, chimpanzees’ populations vary in the diversity and complexity of their foraging behaviours. Whether tolerance is linked to foraging complexity across populations is of course a crucial point, but one which is not yet clear.

Reviewer #2 (Remarks to the Author):

This paper describes how patterns of peering behaviour in immature wild chimpanzees are associated with age, role model and feeding complexity. Overall the results are clearly presented and described and the paper makes a novel contribution to the literature in this area. The figures are clear and helpful in graphically representing some of the key findings. My comments relating to this paper are below.

We thank reviewer 2 for this positive feedback.

Title and abstract

It would be helpful in either the title or abstract to provide the scientific name for this chimpanzee species given patterns of tool use vary between sub-species of chimpanzee.

Thank you for this suggestion, we included it in the abstract.

It would be helpful to provide the age range and sex of the immature chimpanzees in the abstract.

We now include the age in the abstract.

Introduction

The introduction does a good job of reviewing the relevant literature and explaining the importance of peering behaviour in the context of social learning.

We thank you for these positive comments.

Lines 109 – 130 – It would be helpful in this paragraph to explain more explicitly the novelty of the current study as compared to previous studies on peering behaviour in immature wild chimpanzees. Is this the first study to examine this behaviour across the whole range of food types?

This is the first study to date to investigate peering across various feeding behaviours in wild chimpanzees. We added “Whilst previous studies focused on peering in one or few food contexts, here we assessed peering across all contexts in which it was observed.” line 122-123.

Line 133 – It would be helpful to provide the weaning age here to give more specificity to the prediction

We have added this information in line 151.

Line 140 – It is not really clear how this prediction relates to ‘tolerant’ mothers. The tolerance of the mothers towards the infants is not measured or mentioned in either of the predictions.

As the term tolerant mothers might be misleading to the readers, we removed it from the prediction title and added “As mothers were more spatially tolerant than other individuals and might provide favourable learning opportunities” line 156-157.

Although we did not evaluate inter-individual variation in maternal tolerance, our data and the literature suggest that mothers are overall more tolerant models than other individuals. This is supported by figure S2C and S3A, showing that mothers are more likely to allow food transfers and accept explicit solicitations than other individuals. Additionally, time spent in close proximity was higher for mothers (We added a new supplementary figure S4). Although some factors leading to reliance on mothers might be unrelated to social tolerance (i.e. attachment, provisioning, socialization processes), maternal tolerance likely facilitates close observation and opportunities to interact with the items during peering.

Methods

Data collection

It would be good to provide some more specific information on the 35 individuals here or in the SI, including how many hours of observation per individual, over what time period were individuals followed, the mean age of each individual across the samples for that individual and the number of each sex.

Thank you for the suggestion. We now added a table with this information in the supplementary (sex, age, period of observation) Table S1.

It is not clear from this description the time period over which this data was collected i.e. were all observations of a specific individual done when they were the same age, or were observations across multiple years?

The data were collected between January 2021 and April 2023, with a mean observation period of 10.10 +/- 4.51 months per individual. We added the period of data collection in the main text line 175 as well as the number of individuals that were born or stopped being followed during this period. We also added a summary of the individuals included in the study in the supplementary (sex, age, period of observation).

Line 168 It would be good to explain how the dyadic association index was calculated here given there are different ways of calculating this.

Thank you for this suggestion. We now included the calculus in the method line 370. (Time A and B spent in the same party)/(Time A seen in parties 370 + Time B seen in parties - Time A and B spent in the same party).

Data set

Line 179 – When describing the peering events, it would be helpful to explain that these were calculated as peering events per hour of the focal. This is explained in line 230 but may be clearer here. Were all peering events treated the same regardless of duration and was the duration recorded? Presumably peering events could vary in length from >5s to over a minute?

The peering was treated as a duration-less event. In our models including either a Poisson or a negative binomial distribution, the response variable was the raw number of peering events. But we included an offset term to model the rate of peering per focal hour. As we precised in the introduction “Consecutive peering events or explicit begging events were scored as two distinct events if the focal individual changed his activity or if its target started the processing of a new item in the interval between them.” Therefore, peering duration was constrained by role model activity and we expect that the type of information that can be extracted is relatively similar between peering events, regardless of their duration.

Statistical analysis

As some readers may be unfamiliar with Bayesian regression models, it may be good to explain why these were chosen over frequentist approaches for this study. Similarly, for readers unfamiliar with a Bayesian approach, it may be helpful to give a brief explanation of how to interpret the 95% and 89% credible intervals, as these differ from the confidence intervals which readers may be more familiar with.

Thank you for this suggestion, we added a small paragraph in the introduction line 241-244 to precise this point in the method. “For all models, we reported the estimate (mean of the posterior distribution) and credible intervals (CI) at: CI89% and CI95% allowing us to discuss strongly supported and

"weakly" supported effects altogether without having to take a dichotomous approach (effect or no effect). (McElreath 2016)."

Line 212 – give the full name for MCMC chain on first usage

Thank you, we added it to the text.

Models

Line 230 – give full name of glmm on first usage

Thank you, we applied those changes to the text.

Line 326 Control fixed effects

The model numbering here does not align with the paper. For example, on line 329 there is a model 2c but this does not appear in the previous section. Further, there are models 4a and 4b which presumably relate to the models presented in the SI.

Thank you for spotting it. There was a mistake in the model names, we now have corrected this.

There needs to be a more explicit justification for why some control variables were included in some of the models but not others. I'm sure there are good reasons for this based on the other variables included in the models, but it would be good to set these reasons out for the reader.

We added precisions on our choices of control variables in this section. Maternal rank was not included in model 4a and 4b for which mothers were excluded, and not in model 1b for which access to resources was already controlled for. Party size was not included in models 4a and 4b as data were either aggregated over a long period of time, or because only the type of social interaction or the response to solicitation were tested (1a and 1c).

Results

Results tables

It may be clearer to use a decimal point rather than a comma in the Results tables.

We applied those changes to the text.

It could be made clearer in the Table title and/or text what the bold and italics signify i.e. presumably that the credible interval does not cross zero?

We now included "credible interval at 95 and 89% (respectively in bold or italic if they did not cross 0)" in the tables.

It could also be made clearer in the Tables (if this is correct) that the response variable is not the 'raw' number of peering events, but the peering rate per hour of focal observation

Our models include either a Poisson or a negative binomial distribution, with the response variable being the raw number of peering events. But as we included an offset term we are effectively modelling the rate of peering per focal hour.

Line 372 – I may be reading Table 2 or the text incorrectly, but the estimates for extractive foraging in the text do not appear to be consistent with Table 2. In the Table 2, for Model 1b, extractive with tools has an estimate of 0.61, but in the text the estimate for extractive with tools is 1.15? (respectively in bold or italic if they did not cross 0)

This is because the reference level shown in table 2 is extractive foraging (so the other levels of the predictor are evaluated against it). However, in the text we also compare the two other levels, non-extractive foraging and extractive foraging with tools, to each other. We have added the information on these comparisons in Table S3.

Two of the headings in the results section (Early learning and learning of complex tasks rely on tolerant mothers; later learning relies on many tolerant role models) seem more like an interpretation of the function of peering behaviour rather than a summary of the actual results reported. It may be better to have headings for these sections that relate more closely to the response variables in the models (peering behaviour), as learning of food processing skills (e.g. increase in processing efficiency) was not directly assessed in these models Model 2a, Figure 3A

Thank you for this suggestion. We adapted the heading of the result section to summarize the results better.

Is this reduction in proportion of peering directed at the mother a function of reduced proximity to mother and increased proximity to other individuals as the immature chimpanzee ages? So, it is not that the immature chimpanzees are actively ‘seeking out’ opportunities to learn from others, but they spend more time in closer proximity to others and therefore have more opportunity to display peering behaviour.

For example, if the proportion of time the immature chimpanzee spent in close proximity to the mother vs. other individuals was plotted against age, there may be a similar pattern with older individuals spending less time in closer proximity to the mother.

Yes, the proportion of time spent in close proximity to the mother in our dataset decreases during development, suggesting that changes in peering behaviour might be at least partially explained by changes in socialization. Note however, that proportion of time spent in close proximity with other dyads also decreases with age to a lesser extent, suggesting that immatures already have opportunities to peer at other role models early in development. In 2A and 2B, the mother is always in the same party composition, so they still have the possibility to peer at their mother in those contexts. Additionally, the fact that role model choices seem influenced by task complexity in 2B suggests that role model choices are not solely explained by proximity.

Model 2b

The key prediction for this model as set out in lines 144-145 and Table S2 is that ‘Mothers will be more likely to be observed for complex tasks’ (quote from Table 2a).

I may be mistaken, but the key test of this prediction would then seem to be the estimates for the variables ‘Non extractive’ and ‘extractive with tools’ in Model 2b i.e. is the identity of the receiver of the peering related to the complexity of the food processing. The credible intervals for both of these variables cross zero.

We think this model supports our prediction. While when the mother is feeding, other individuals were feeding in 90% of the cases (see intro statistical analysis line 320), the probability of having good alternative role models likely increases with party-size. This is supported by the fact that almost all peering events were directed at the mother for small party sizes, regardless of the type of foraging. We therefore expect a stronger effect of task complexity for large party-sizes. Because of this, we considered that showing the two-way interaction between the task complexity and party-size would be more relevant to our prediction. Note however that when the levels, non-extractive foraging and extractive foraging with tools were compared (for a mean party-size), we obtained a positive effect of tool use on the probability to peer at the mother at 89%CI (this is not shown in the table because the reference category per default is extractive foraging without tools, but we added it in the supplementary for the new version). We now added more details on our prediction (lines 338-339) “Given that the probability of having suitable role models should increase with party-size, we included a two way interaction between task complexity and party-size to test if biases toward the mother persist when party size increases.”, and discussed average effects in the result section (line 503).

It should be made clear in the text whether the key prediction as set out in lines 144-145 and Table S2 is supported by model 2b or not. This also applies to the statements relating the task complexity and reliance on mother in the discussion.

Instead in the text (lines 414 – 417) and Figure 3b, the reported results relate to the interaction between role food processing complexity and party size. Whilst this is interesting, overall model 2b does not seem to support the core prediction that ‘Mothers will be more likely to be observed for complex tasks’ as in Table S2 and lines 144-145. This prediction related to the ‘main effect’ of task complexity, not the interaction between task complexity and party size.

There is also some inconsistency in the numbering of the models (2b or 3b) in this section in the text and table.

Thank you, we corrected this in this section as well.

Table 4 - Dyadic association index is listed twice in Model 3b with different estimates.

The second occurrence corresponded to a two-way interaction between age and DAI. We corrected this mistake in the table.

Fig 4 C and D – should second value on Y axis be 0.001?

Yes, we corrected it on the plot.

Discussion

Line 472 – is the term caregiver here used instead of mother? As mother was used throughout the manuscript, it may be clearer to use this term as well here or explain what is meant by the term caregiver (i.e. does this include caregivers other than the mother).

Yes, the term mother is more relevant here. We changed it in the discussion line 556.

Line 529 – is this statement supported by the model results? See comments on model 2b above.

We think this statement is supported by our results as discussed above.

Line 584 – I am not sure whether the results support the statement relating to high maternal tolerance, as maternal tolerance was not directly measured in this study as far as I can tell.

Although we did not tease apart individual variations in maternal tolerance, we assumed that mothers were more tolerant than other role models during learning of feeding skills as suggested by the literature in chimpanzees (see: Boesch et al. 2019; Fröhlich et al. 2020 for studies conducted in Tai). Our models supported this assumption, as mothers were more likely to allow food transfers (see figure S2C). We further confirmed this by conducting a dyadic-level analysis to compare the probability of being in proximity to the mother in comparison to other individuals (see figure S3).

Although factors leading to reliance on mothers might be unrelated to social tolerance (i.e. attachment, provisioning, socialization processes), one of the outcomes of peering at mothers compared to other role models is to receive higher tolerance.

Overall comments

The finding that peering appears to function to acquire information more than food from others is an important one and is referenced in the abstract and the first subheading in the Discussion. But the analysis for this is presented in the SI. Unless there are word limits or space constraints, I would suggest moving this analysis to the main paper as it is only three additional models. In this case the results would have to be explained more fully as for the results in the main paper.

Thank you for this suggestion, we therefore included the analysis in the main text and changed the introduction, method, results and discussion sections accordingly. whilst other changes are in red, we mark the parts moved from the SI in blue.

As part of the review, I am asked to comment on ‘the ability of a researcher to reproduce the work, given the level of detail provided’. The models are clearly described, but I could not see a statement about availability of data and analysis code. It would greatly aid a researcher to reproduce the work if the data and analysis code were provided. Given there are no issues around anonymity and confidentiality as there sometimes are with human data, I see no reason why the data and analysis code should not be provided in line with open science principles.

We did not include a data availability section at the first submission. We now added this section in the manuscript highlighting that the data and the code will be available with the paper.

References

Leeuwen, Edwin J. C. van, et William Hoppitt. 2023. « Biased cultural transmission of a social custom in chimpanzees ». *Science Advances* 9 (7): eade5675. <https://doi.org/10.1126/sciadv.ade5675>.

Boesch, Christophe, Daša Bombjaková, Amelia Meier, et Roger Mundry. 2019. « Learning Curves and Teaching When Acquiring Nut-Cracking in Humans and Chimpanzees ». *Scientific Reports* 9 (1): 1515. <https://doi.org/10.1038/s41598-018-38392-8>.

Estienne, Vittoria, Heather Cohen, Roman M. Wittig, et Christophe Boesch. 2019. « Maternal Influence on the Development of Nut-cracking Skills in the Chimpanzees of the Tai Forest, Côte d’Ivoire (*Pan Troglodytes Verus*) ». *American Journal of Primatology* 81 (7).

<https://doi.org/10.1002/ajp.23022>.

Berdugo, Sophie, Emma Cohen, Arran J. Davis, Tetsuro Matsuzawa, et Susana Carvalho. 2024. « The Ontogeny of Chimpanzee Technological Efficiency ».

Lonsdorf, Elizabeth V. 2006. « What Is the Role of Mothers in the Acquisition of Termite-Fishing Behaviors in Wild Chimpanzees (*Pan Troglodytes Schweinfurthii*)? » *Animal Cognition* 9(1):36-46. doi: 10.1007/s10071-005-0002-7.

We are pleased to resubmit our revised manuscript to *Communications Biology*. We have thoroughly addressed all comments and integrated the revisions (highlighted in red). Below, we detail our responses and modifications.

REVIEWERS' COMMENTS:

Reviewer #1 (Remarks to the Author):

The authors have done a superb and thorough job responding to reviewer feedback. The flow of the Introduction is much improved, some details in the Methods have been clarified, and the figures are now easier to interpret. Overall, the manuscript is well-written, the analyses are robust, and the discussion is cogent and insightful. I have very few remaining minor comments, which mostly focus on the organization of the Introduction.

Thank you very much for this positive feedback.

MINOR COMMENTS

L27: Remove the “the” to keep it parallel with 3).

Thank you for this suggestion, we removed it from the text.

L46-68: Maybe I’m misunderstanding, but I’m still not following the flow of the argument here and the introduction of social tolerance merged with the discussion of caregivers. Are there species that might lack toolkits because their caregivers (e.g., mothers) are so rejecting that offspring cannot observe their mothers’ feeding behavior and strategies? More broadly, I think the Introduction would be more clear to me if the text and references in L47-56 were placed after the discussion of caregivers (i.e., L56-68) and paired with the paragraph discussing the wider pool of role models (L69-83).

Our main argument in this paragraph is that social tolerance is needed to acquire complex skills. Here, caregivers allow us to test this idea because caregiver-offspring exhibit the highest levels of tolerance within social groups (**Boesch et al. 2019**). In this line, between populations variations in maternal tolerance might also correlate with complexity of tool use behaviour in chimpanzees (**Musgrave et al. 2020**).

In the next paragraph we want to develop the idea that in addition to social tolerance, the number and the diversity of role models is important to increase and diversify learning opportunities.

To clarify this point we have changed the sentence:” In various species, immatures receive a high level of social tolerance from their caregiver and use them as primary role models.”, to “**Across species, caregivers demonstrate greater levels of social tolerance compared to other individuals and often serve as primary role models**”.

L84: “on” should be “in”

Thank you for pointing this out, we have changed it accordingly.

L146-147: I’m curious about the logic behind this hypothesis. Might it also be possible to expect offspring to engage in more explicit begging if the task is difficult for them to accomplish on their own? Or is this unlikely since the tasks are impossible without a decent mastery of the tools themselves?

“1c) Processing complexity will increase the relative use of peering behaviour compared to explicit begging behaviour.”

Here, we expect that the primary motivation for immature individuals to peer or beg in a feeding context is to obtain resources that they cannot otherwise access. This may occur for one of two main reasons:

(1) the resource is difficult to obtain because it is rare or monopolized by other individuals, or (2) immatures are either incapable of processing it or not efficient enough, which is more commonly the case in extractive foraging and tool use (e.g., immatures are not able to crack nut successfully before the age of 4).

In the second scenario, but not the first, we expect that gaining information about the processing steps will be beneficial for the immature. Therefore, peering and begging should both increase with task complexity, but peering should be used proportionally more for complex tasks.

We added more information on this prediction in the method: “**Immatures mainly interact in feeding contexts to access resources they cannot obtain alone. This may be because the resource is rare or monopolized or because they lack the skill to process it. In the latter case, gaining information on processing is beneficial. Thus, if peering provides information, it should increase in proportion with task complexity.**”

Figure 4b, 4c: Can the scales be the same for the adjacent figures?

Yes, we changed the scale of the plot.

Reviewer #2 (Remarks to the Author):

The authors have done a very good job in responding to the reviewer comments. They have fully addressed all the points I made in the review and made their data and code available.

Thank you very much for this positive comment.

References

Boesch, C., Bombjaková, D., Meier, A. & Mundry, R. Learning curves and teaching when acquiring nut-cracking in humans and chimpanzees. *Sci Rep* **9**, 1515 (2019).

Musgrave, S. *et al.* Teaching varies with task complexity in wild chimpanzees. *Proc Natl Acad Sci U S A* **117**, 969–976 (2020).